# Locality-Aware Multiresolution Graph Filtering to Mitigate Oversmoothing and Oversquashing

## Abstract

Real-world graphs demonstrate region-specific heterophily, where some regions are smooth and suitable for low-pass averaging and others are sharp and necessitate high-pass contrast. However, spectral GNNs that utilize a global eigenbasis or high-order polynomial surrogates for filtering rely upon dense, non-orthogonal, non-local bases that blend signals from distant, semantically unrelated regions. This results in small clusters connected to large homogeneous hubs becoming indistinguishable and overshadowed by continuous global mixing, exacerbating oversmoothing. To address these limitations, this paper presents Hierarchical Spectral Learning (HSL), a locality-first, multiresolution framework for learning frequency signal regions by region. HSL estimates diffusion neighborhoods and constructs a diffusion barrier to identify incompatible neighbors that are weakly connected under diffusion (i.e., connected by long, low-weight paths), preventing incompatible clusters from being merged together. A lightweight MLP layer is applied on the bottom-K eigenspace of a multi-view Laplacian (which combines topology and feature affinity), generating soft node scores for coarsening the graph into a hierarchy. Rather than a full graph-wide eigen decomposition, at each level, we create a strictly local, degree-aware block-sparse orthonormal Haar basis that consists of a low-pass scaling wavelets and a high-pass inter-/intra-cluster wavelets. The model has linear complexity operations per layer while preserving strict locality. The multi-scale unpooling layer combines coarse and fine signals to preserve small, high-contrast local structures while maintaining global context. The hierarchy reduces effective path lengths for message passing, preventing oversquashing. Empirically, HSL attains state-of-the-art performance with linear scalability, enhancing node classification by up to $3\%$ on heterophilous benchmarks and up to $7\%$ on graph classification tasks.

## 1 Introduction

Real-world graphs showcase regional heterogeneous properties, where some regions exhibit varying intensities of similar graph topology and node feature similarity. In graph learning, low-variation regions (higher similarity in features and topology) benefit from averaging (low-pass filtering) the signals, whereas boundaries between semantically distinct (lower similarity in features and topology) communities require contrast amplification (high-pass filtering) (Chien et al., 2020). Mainstream spectral approaches implement this filtering by projecting features onto a global eigenbasis or its polynomial approximation (Zhu & Koniusz, 2021). Many "heterophily-aware" spatial methods similarly learn propagation operators that eventually induce a global basis (Chien et al., 2020). This suboptimal global basis exhibits several fundamental limitations, as discussed by He et al. (2022) and its high-order global polynomial approximation by Huang et al. (2024): (i) *Non-locality* — the frequency energy of a node spreads across multiple hop distances, disregarding locality; (ii) *Non-orthogonality* — substantial inter-basis correlations cause energy associated with one frequency band to leak into others, thereby reducing local signal preservation; and (iii) *Density of the basis*, which increases computation and memory requirements, thus limiting scalability on large graphs.

Moreover, the construction of the global basis relies solely on topology and remains oblivious to the feature similarity across different regions of the graph (Zheng et al., 2024). As a result, these

methods blend information from distant, semantically unrelated areas of the graph without knowing the comprehensive characteristics, degrading the ability to preserve fine-grained local structure (Li et al., 2023). Further, this global signal mixing is magnified in graphs dominated by large homogeneous clusters (hubs), where small homogeneous clusters (spokes) are densely connected to hubs. Repeated propagation of the global mixing algorithm enables the hub signals to dominate the spoke signals, thereby mitigating the well-known challenge of oversmoothing (Fesser & Weber, 2024). Additionally, increased depth of message propagation hinders information from reaching the farthest semantically aligned neighbors, causing oversquashing (Fesser & Weber, 2024). To address these challenges, a related work (Anonymous, 2025) proposed a hierarchical pooling method that utilizes sparse Haar bases with GNN Encoder-guided clustering. However, its reliance only on graph topology and potentially noisy encoder signals results in suboptimal clustering outcomes.

Motivated by the above limitations, we propose a locality-first, multiresolution framework named Hierarchical Spectral Learning (HSL) that builds a robust multiscale spectral filter. HSL first estimates diffusion neighborhoods and constructs a diffusion barrier that enforces short-path locality by marking pairs that are diffusion-distant as incompatible. Using this barrier, we compute the bottom K eigenspace as a spectral embedding and feed it to an MLP head to predict soft cluster indicators. Under mild diffusion node cohesion and eigen energy gap conditions, this eigenspace coincides with the ground-truth cluster indicator subspace. Repeating this coarsening algorithm, we obtain a multiresolution hierarchy tree. Then, at each level of the tree, we construct a strictly local, sparse, orthogonal, and degree-aware Haar basis comprising (i) a scaling vector (local mean), (ii) inter-cluster contrasts, and (iii) in-cluster contrasts. Each level's Haar matrix basis is block-sparse with $O(|V^\ell|)$ nonzero entries where $V^\ell$ is the number of nodes at level $\ell$. The construction of the hierarchy and its sparse basis incurs only linear computational cost, while preserving locality and preventing global signal mixing. We then apply diagonal spectral filtering in this basis, with learned band gains that amplify high-frequency content at heterophilous boundaries and apply low-pass smoothing in homogeneous regions. Finally, we fuse features across coarse and fine levels to retain global context while preserving small, high-contrast structures. By separating hubs and spokes during coarsening, HSL learns their contrast rather than smoothing spokes into hubs, thereby mitigating hub domination and oversmoothing. The coarsening hierarchy reduces effective path lengths logarithmically, allowing gradients and information to propagate without the exponential decay, which alleviates oversquashing (Di Giovanni et al., 2023).

Our main contributions are as follows: (I) We introduce a novel framework, HSL, that constructs a locality-aware, orthonormal, multi-scale, and sparse basis from a multi-view Laplacian (topology + feature affinity), enabling linear construction and filtering on large graphs. HSL learns graph signals across multiple resolutions while explicitly preventing global signal mixing. (II) We rigorously prove and empirically confirm that conventional GNNs suffer from hub domination, oversmoothing, and oversquashing, whereas our hierarchical local Haar construction and adaptive learning of low-pass and high-pass filtering gains avoid all the pathologies. (III) We demonstrate that HSL achieves state-of-the-art accuracy on both node and graph classification tasks while maintaining linear scalability.

## 2 RELATED WORKS

**Spectral Filtering:** Early models, like ChebNet (He et al., 2021), used truncated Chebyshev polynomials to approximate the eigenbasis. Conversely, Generalized Page Rank GNN (GPR-GNN) (Chien et al., 2020) utilizes generalized PageRank weights with monomial bases for approximations. In contrast, BernNet (He et al., 2021) and JacobiConv (Wang & Zhang, 2022) provide Bernstein and Jacobi polynomials, respectively, to enhance the interpretability and adaptability of the bases. ChebNetII (He et al., 2022) addressed the issue of Chebyshev polynomials overfitting by employing interpolation, whereas OptBasisGNN (Guo & Wei, 2023) aimed to make basis polynomials orthogonal to accelerate convergence. Despite these developments, many of these methodologies continue to employ dense, fixed graph-wide bases that prevent global mixing of messages.

**Hierarchical Graph Learning:** Learnable pooling methods, such as DiffPool (Ying et al., 2018), SAGPool (Lee et al., 2019), TopKPool (Diehl, 2019), and gpool (Gao & Ji, 2019), utilize node-scoring or soft assignments to coarsen homophilous graphs at a singular resolution. These methods incur an $O(n^2)$ computational cost (Li et al., 2024). EigenPool (Ma et al., 2019) and Haar-

based pooling employ an expensive eigendecomposition to coarsen the graph, incurring an $O(n^3)$ computational cost, which constrains scalability and overlooks local heterophily (Wang et al., 2020).

## 2.1 PRELIMINARIES

Let $G = (V, E)$ be a simple, undirected graph, where $V$ denotes the set of vertices and $E$ denotes the set of edges. Its adjacency matrix $A \in \mathbb{R}^{m \times m}$ has entries $A_{ij} = 1$ if $i, j \in E$ and 0 otherwise, and $D = \text{diag}(A\mathbf{1})$ is the degree matrix, where $\mathbf{1}$ is the all-ones vector. Given node features $X \in \mathbb{R}^{m \times d}$ and normalized Laplacian $L = U\Lambda U^\top$, a spectral layer rescales graph Fourier components by $g(L)X = U\,g(\Lambda)\,U^\top X$, and in ploynomial spectral methods implements $g$ with a polynomial, $g(L) \approx \sum_{r=0}^{R} \theta_r\,L^r$, where $R \in \mathbb{N}$ is the polynomial order. With $k$ layer message–passing, a GNN learns the graph signal as a polynomial in a symmetric propagation operator $P$, which essentially induces the global eigenbasis of the propagation operator $P$ given by

$$H^{(k)} \approx \sum_{r=0}^{k} \Theta_r\,P^r\,X = g_P(P)\,X = U_P\,g_P(\Lambda_P)\,U_P^\top X, \tag{1}$$

where $U_P \Lambda_P U_P^\top$ is the eigendecomposition of $P$ and $\Theta_r$ are learnable linear maps.

Real-world graphs often exhibit different levels of homophily in different regions of the graph. We quantify *homophily* as $H_{\text{lab}} = \frac{1}{|E|} \sum_{(u,v) \in E} [y_u = y_v]$, where $y$ is the node label. To address the discrepancy between nodes homophily, the *sign- based message passing (SMP)* (Liang et al., 2024; Chien et al., 2020) algorithms introduced signed adjacency $S$ matrix defined as $S_{uv} = +1$ if $(u, v) \in E$ and $y_u = y_v$, $S_{uv} = -1$ if $(u, v) \in E$ and $y_u \neq y_v$ . So the feature of each layer is updated as, $H^{(k+1)} = \sigma\big(S\,H^{(k)}W^{(k)}\big)$, $H^{(0)} = X$, where $S$ works as a propagation operator and $W$ is weight matrix. Using equation (1), the k-layer SMP can be expressed as a spectral filter in the global basis of $S$. To solve the sign flipping of SMP, *chunked multi-track aggregation (CMA)-based methods* (Pei et al., 2024) propose to split each node's neighbors into $t$ tracks (e.g., homo/hetero) with learned attention $s_{ij,t}$ and aggregate messages $m_{i,t}$ per track $t$ given as,

$$m_{i,t} = \sum_{j \in \mathcal{N}(i)} s_{ij,t}\,\big(H_j^{(k)}W_t\big), \qquad H_i^{(k+1)} = \sigma\Big((1-\beta)\,H_i^{(0)}W_0 + \beta \sum_{t=1}^{C} m_{i,t}\Big). \tag{2}$$

where $\sigma$ denotes sigmoid function and $\beta > 0$. The existing methods incur these two fundamental limitations. *(i) Oversmoothing* occurs as the number of layers grows and all node embeddings converge to their class means (Epping et al., 2024). *(ii) Oversquashing* (Topping et al., 2021), which occurs when gradients (or messages) from distant nodes decay exponentially with graph distance, preventing long-range information flow.

## 3 LIMITATIONS OF EXISTING MODELS

The SMP (Liang et al., 2024) and CMA (Pei et al., 2024) methods, discussed above, have the following fundamental limitations: **(I) Hub Dominiatoin:** In many real-world graphs, it is often seen that two small clusters $A$ and $B$, with sizes $|A| = a$ and $|B| = b$ respectively, are each tightly interconnected with a large, homophilous hub $H$ of size $|H| = M$ (where $a, b \ll M$). Although nodes in $A$ and $B$ are categorized differently, both SMP and CMA/M2M require passing messages primarily through the same hub signals, as illustrated in Figure 1. The hub dominates the spokes' messages, leading to their final attributes of all the nodes blending into an indistinguishable embedding space.

**(II) Suboptimal Basis** Spectral approaches rely on fixed eigendocpositions or polynomial bases (e.g., Bernoulli, Chebyshev), which have three major limitations. First, they create dense, globally supported vectors with nonzero weights distributed over almost all nodes. Therefore, perturbing the signal of a single node affects the basis functions across the whole graph. Second, these polynomial

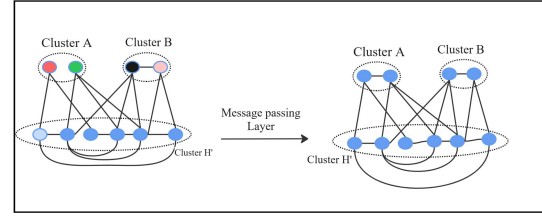

Figure 1: Hub domination in cluster $(A, B, H')$

bases are non-orthogonal across the nodes
and have a strong inter-basis correlation, allowing filtering in one frequency band to leak into others, which exacerbates hub domination effects. Finally, the dense basis renders the model unscalable for larger graphs due to higher computation. Theorem 1 formalizes the claim I.

**Theorem 1** *[Exponential collapse under global-basis models] For architectures based on SMP, CMA, or spectral filters defined in a fixed global basis, the representational distance between any node in A and any node in B decays exponentially with depth. Consequently, after a sufficiently large number of layers, the two become indistinguishable. (Proof in Appendix A.1)*

In addition to analytical results in Theorem 1, using synthetic heterophillous graph datasets, we empirically validated the hub domination phenomenon, as shown in Figure 4.

## 4 METHODOLOGY

In this section, we describe the formulation of the diffusion heat kernel-guided hierarchical tree, followed by the basis formulation for each of the three levels. We then describe the filtering process and provide a rigorous theoretical guarantee that the proposed HSL effectively solves hub domination, oversmoothing, and oversquashing.

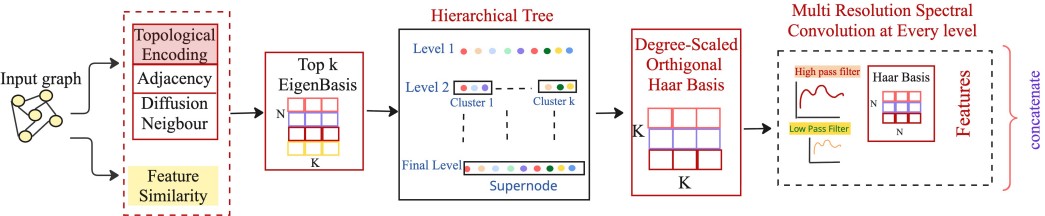

Figure 2: Overview of the proposed HSL framework. The process begins with hierarchical graph coarsening guided by Topology and Feature similarity. At each level, orthogonal, local bases are constructed, followed by diagonal spectral filtering. Finally, multi-scale fusion aggregates transformed messages from all levels.

### 4.1 MULTI VIEW LAPLACIAN AND DIFFUSION DISTANCE BALLS

Since feature similarity between nodes or regions in the graph carries profound information about the graph characteristics, we need to take the feature similarity into account (Zheng, 2024). So, instead of relying exclusively on the graph topology to construct the Laplacian, we have incorporated feature affinity as an additional insight. The feature affinity Laplacian is defined as $L^{(2)} = I - D_{\text{feat}}^{-1/2} W_{\text{feat}} D_{\text{feat}}^{-1/2}$, where $D_{\text{feat}} = \text{diag}(W_{\text{feat}})$. To ensure scalability for a larger graph, the feature-affinity weight matrix $W_{\text{feat}}$ is computed from a $k$-nearest neighbor (k-NN) using FAISS (Danopoulos et al., 2019) algorithm, given by

$$(W_{ij})_{feat} = \begin{cases} \exp\left(-\frac{\|x_i - x_j\|^2}{2\sigma^2}\right), & j \in N_k(i), \\ 0, & \text{otherwise,} \end{cases} \tag{3}$$

where $N_k(i)$ denotes the $k$ nearest neighbors of node $i$. At each hierarchy level $\ell$, for an active cluster $C$ (or the full graph at the root), we form a data-adapted mixture of two Laplacian views as follows

$$L^{\text{mix},C} = \sum_{k=1}^{2} \alpha_C^{(k)} L^{(k)}, \quad \alpha_C = \text{softmax}\big(g_\theta(\phi_C)\big), \tag{4}$$

where $g_\theta$ is a lightweight network fed with cluster statistics $\phi_C$ (e.g., conductance, and local homophily ratio). Conductance measures the proportion of a cluster's total edges that go to nodes outside the cluster. The $L^{\text{mix},C}$ ensures that both topology structure features and affinity between nodes influence the clustering process.

To avoid clustering nodes that are structurally dissimilar and superficially related, we determine locality within each cluster $C$ with the heat kernel $H_{t,C} = \exp\big(-t\,L_{\text{mix},C}\big)$, where $t$ controls the locality scale (Thanou et al., 2017). It averages information along many short paths while suppressing weak, long-range connections. The diffusion distance between nodes $i$ and $j$ is measured as $d_t(i,j) = \big\|e_i^\top H_{t,C} - e_j^\top H_{t,C}\big\|_2$, where $e_i \in \mathbb{R}^{|C|}$ denote the $i$-th standard basis vector (1 at index $i$, 0 elsewhere). For each node $i$, the *diffusion ball* at scale $t$ is $B_t(i,\varepsilon) = \{\, j \in C \, : \, d_t(i,j) \leq \varepsilon \,\}$, and nodes outside $B_t(i,\varepsilon)$ are treated as diffusion-distant relative to $i$. We encode these diffusion distant nodes in a sparse, symmetric *incompatibility matrix* as

$$(M_C)_{ij} = \frac{\mathbf{1}\{\, j \notin B_t(i,\varepsilon) \,\}}{D_i D_j}. \tag{5}$$

where $D$ denotes the degree. We approximate the heat kernel using polynomials (details are provided in Appendix B.1). In the next section, we describe how both $L_{mix}$ and $M_C$ guide the construction of a hierarchical tree.

## 4.2 Formulating Hierarchical Tree and orthogonal basis

We build a hierarchical tree by recursively partitioning each cluster into balanced subclusters. For each cluster $C$, we combine $L_{\text{mix},C}$ and $(M_C)_{i,j}$ in a single positive semi-definite (PSD) operator,

$$L_C = L^{\text{mix},C} + \lambda M_C, \quad \lambda \geq 0. \tag{6}$$

We seek a $K$-dimensional spectral embedding by solving the balanced relaxation as

$$\min_{U \in \mathbb{R}^{|C| \times K}} \text{Tr}(U^\top L_C U) \quad \text{s.t.} \quad U^\top U = I_K, \;\; U^\top s_C^{(0)} = 0, \tag{7}$$

where $s_C^{(0)} = \frac{D_C^{1/2}\mathbf{1}}{\|D_C^{1/2}\mathbf{1}\|_2}$ and $D_C$ is diagonal matrix of the cluster. The solution is given by the $K$ bottom *nontrivial* eigenvectors of $L_C$, which we obtain via power iteration (Panju, 2011). Minimizing $\text{Tr}(U^\top L_C U)$ drives the minimizer $U$ into an embedding space that is smooth under the mixed geometry, while the penalty term $\lambda$ on $M_C$ discourages clustering nodes that are diffusion-distant, countering hub dominations. The balance constraint $U^\top s_C^{(0)} = 0$ enforces degree-balanced partitions, ruling out trivial solutions that concentrate all degree mass on one side. This process encourages nodes that are similar in mixed geometry to lie near each other in $U$-space, while heterophilous or diffusion-distant pairs are pushed far apart. Theorem 2 below guarantees that a choice of $\lambda$ ensures no diffusion-distant pairs are co-clustered.

**Theorem 2** *Let $S^\star = \{S_1, \ldots, S_K\}$ be a degree-balanced ground-truth partition on $C$, and let $Y^\star \in \mathbb{R}^{|C| \times K}$ denote its degree-normalized indicator matrix. Assume: (i) diffusion cohesion: if $i, j \in S_k$ then $j \in B_t(i,\varepsilon)$; (ii) diffusion margin: if $i \in S_k$, $j \in S_\ell$, $k \neq \ell$, then $j \notin B_t(i,\varepsilon)$; and (iii) geometry gap: for any degree-balanced merge $S$ across parts, $\Phi_{\text{mix}}(S) - \Phi_{\text{mix}}(S^\star) \geq \Delta_{\text{mix}} > 0$, where $\Phi_{\text{mix}}(S) = \text{Tr}(Y^\top L_{\text{mix},C} Y)$ for the indicator $Y$ of $S$. Let $\Psi_{\min} = \text{Tr}(Y^\top M_C Y) > 0$ over the same family of balanced merges $S$. If $\lambda \geq \Delta_{\text{mix}}/\Psi_{\min}$ the bottom-$K$ eigenspace of $L_{\text{mix},C} + \lambda M_C$ equals $\text{span}(Y^\star)$. Consequently, balanced rounding (or a linear/MLP head applied to $U$) recovers $S^\star$ exactly; no diffusion-distant pairs are co-clustered.(Proof in Appendix A.2)*

Given $U$ at level $\ell$, the $i$ th row $N_i^{(\ell)} = U_{i:}$ serves as as the node coordinate of $i$. To resolve the rotational ambiguity of $U$, we score cluster affinities with a linear head as

$$\alpha_i^{(\ell)} = W^{(\ell)} N_i^{(\ell)} + b^{(\ell)} \in \mathbb{R}^{K_\ell}, \quad \alpha_{ik}^{(\ell)} = \big(\alpha_i^{(\ell)}\big)_k. \tag{8}$$

**Margin adjustment and Soft assignments.** To capture uncertainty near decision boundaries, we compute a normalized margin $\mu_i^{(\ell)}$ between the top two scores as

$$\mu_i^{(\ell)} = \frac{\alpha_{i,\max}^{(\ell)} - \alpha_{i,2nd}^{(\ell)}}{|\alpha_{i,\max}^{(\ell)}| + |\alpha_{i,2nd}^{(\ell)}| + \zeta}, \quad \zeta = 10^{-3}. \tag{8}$$

A small $\mu_i^{(\ell)}$ indicates ambiguity (node lies near a boundary), while a large $\mu_i^{(\ell)}$ indicates confident separation. We obtain probabilistic memberships of the cluster, denoted as soft-assignment matrix $A_s^{(\ell)}$, by applying a margin-scaled softmax as

$$\big(A_s^{(\ell)}\big)_{ik} = \text{softmax}_k\big(\mu_i^{(\ell)} \alpha_{ik}^{(\ell)}\big), \quad A_s^{(\ell)} \in \mathbb{R}^{|V^{(\ell)}| \times K_\ell}. \tag{9}$$

**Feature and edge aggregation.** Let $X^{(\ell)} \in \mathbb{R}^{|V^{(\ell)}| \times d_\ell}$ and $A^{(\ell)} \in \mathbb{R}^{|V^{(\ell)}| \times |V^{(\ell)}|}$ denote features and adjacency at level $\ell$. We coarsen by probability-weighted aggregation given by

$$X^{(\ell+1)} = \left(A_s^{(\ell)}\right)^\top X^{(\ell)} \in \mathbb{R}^{K_\ell \times d_\ell}, \qquad A^{(\ell+1)} = \left(A_s^{(\ell)}\right)^\top A^{(\ell)} A_s^{(\ell)} \in \mathbb{R}^{K_\ell \times K_\ell}. \quad (10)$$

Nonzero entries of $A_s^{(\ell)}$, i.e., $(A_s^{(\ell)})_{ik} > 0$, establish parent–child relationships, resulting in the hierarchical framework employed in the subsequent Haar construction. So, the feature of each coarse node is the sum of the features of its children, weighted by their probabilities. The coarse adjacency is the sum of all the fine-level connections between the child sets. The coarsening loop continues until the size of the coarse graph meets the configurable limit $h$: $|V^{(L)}| \leq h$. Then, the tree is used to determine the class-aware multiway Haar basis $U^{(\ell)}$ at each level.

**Basis Formulation.** Since the coarsest (last) layer collapses all nodes into a single cluster, we use a bottom-up scheme. We compute the Haar basis at the coarsest layer and then recursively lift it to obtain bases for all finer (upper) levels. The details are given below.

**A. Coarsest level.** At the coarsest level $L$ let a graph/cluster consist of $K_L = |V^{(L)}|$ supernodes, The basis $U^{(L)} \in \mathbb{R}^{K_L \times K_L}$ is calculated in two steps:

**(i) Global Scaling Vector (degree-aware).** The first basis vector is the degree-weighted constant $u_{\mathrm{sc}}^{(L)} = \frac{D^{(L)\,1/2}\mathbf{1}}{\left\| D^{(L)\,1/2}\mathbf{1} \right\|_2}$, which captures the low-frequency (average) component across all $K_L$ nodes.

**(ii) Orthonormal Wavelets (degree-aware).** Index the supernodes by $i = 1, 2, \ldots, K_L$ and let $v_i = d_i^{(L)}$ be their degree masses. For each wavelet $q = 1, 2, \ldots, K_L - 1$, define prefix/suffix degree masses $V_A(q) = \sum_{i=1}^q v_i, V_B(q) = \sum_{i=q+1}^{K_L} v_i$. Define the $q$-th Haar wavelet $w_q^{(L)} \in \mathbb{R}^{K_L}$ by

$$w_q^{(L)}(i) = \begin{cases} \sqrt{\frac{V_B(q)}{V_A(q)(V_A(q)+V_B(q))}}, & 1 \leq i \leq q, \\ -\sqrt{\frac{V_A(q)}{V_B(q)(V_A(q)+V_B(q))}}, & q+1 \leq i \leq K_L. \end{cases} \quad (11)$$

Now it is easily realizable that $\sum_{i=1}^{K_L} w_p^{(L)}(i) = 0$, $\|w_q^{(L)}\|_2 = 1$, and $\langle w_p^{(L)}, w_q^{(L)} \rangle = 0$ for $p \neq q$.

**B. Finer levels.** Next, for each finer level $\ell = L-1, L-2, \ldots, 0$, we build the basis $U^{(\ell)}$ in three steps:

**(i) Propagate the Scaling Vector.** Given the coarser scaling vector $u_{\mathrm{sc}}^{(\ell+1)} \in \mathbb{R}^{|V^{(\ell+1)}|}$ and the soft-assignment matrix $A_s^{(\ell)} \in \mathbb{R}^{|V^{(\ell)}| \times |V^{(\ell+1)}|}$, we lift the constant component as follows:

$$u_{\mathrm{sc}}^{(\ell)} = A_s^{(\ell)} u_{\mathrm{sc}}^{(\ell+1)} \in \mathbb{R}^{|V^{(\ell)}|}, \qquad u_{\mathrm{sc}}^{(\ell)} \leftarrow \frac{u_{\mathrm{sc}}^{(\ell)}}{\|u_{\mathrm{sc}}^{(\ell)}\|_2}. \quad (12)$$

**(ii) Intra-Cluster Wavelets (between clusters).** At level $\ell$, let the graph be partitioned into $K_\ell$ clusters $C_1^{(\ell)}, C_2^{(\ell)}, \ldots, C_{K_\ell}^{(\ell)}$. Let the cluster degree masses be $v_k^{(\ell)} = \sum_i d_i^{(\ell)} s_{i,k}^{(\ell)}$. Wavelets between these nodes in one cluster are calculated according to equation (11).

**(iii) In-Cluster Wavelets (inside a cluster).** Let $C_k^{(\ell)}$ be a cluster at level $\ell$ with membership vector $s_k^{(\ell)} = (A_s^{(\ell)})_{:,k} \in \mathbb{R}^{|V^{(\ell)}|}$. Label its $n_k$ nodes by $i_{k,1}, \ldots, i_{k,n_k}$. For each split $r = 1, \ldots, n_k - 1$, define the degree-weighted masses $\alpha_{k,r} = \sum_{j=1}^r d_{i_{k,j}}^{(\ell)} s_{i_{k,j},k}^{(\ell)}, \beta_{k,r} = \sum_{j=r+1}^{n_k} d_{i_{k,j}}^{(\ell)} s_{i_{k,j},k}^{(\ell)}$. Then the in-cluster wavelet $w_{k,r}^{(\ell)} \in \mathbb{R}^{|V^{(\ell)}|}$ is

$$w_{k,r}^{(\ell)}(i_{k,j}) = \begin{cases} \frac{\beta_{k,r}}{\alpha_{k,r} + \beta_{k,r}} s_{i_{k,j},k}^{(\ell)}, & j \leq r, \\ -\frac{\alpha_{k,r}}{\alpha_{k,r} + \beta_{k,r}} s_{i_{k,j},k}^{(\ell)}, & j > r, \end{cases} \quad \text{and } w_{k,r}^{(\ell)}(i) = 0 \text{ for } i \notin C_k^{(\ell)}. \quad (13)$$

Each such wavelet has zero mean on its cluster, unit norm, and is orthogonal.

Stacking the scaling vector and all in-cluster wavelets gives the orthonormal basis

$$U^{(\ell)} = \left[ u_{\mathrm{sc}}^{(\ell)} \,\|\, W_{\mathrm{inter}}^{(\ell)} \,\|\, W_{\mathrm{intra}}^{(\ell)} \right] \in \mathbb{R}^{|V^{(\ell)}| \times |V^{(\ell)}|}. \quad (14)$$

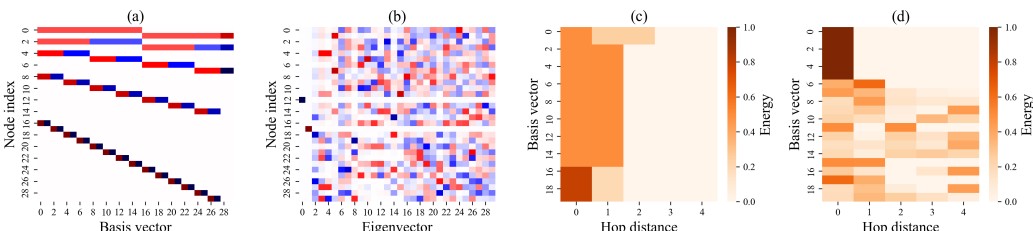

Figure 3: (a) **Signed Haar Basis**. On a 28-node synthetic graph, each column represents a sparse, highly localized wavelet with coefficients concentrated on a small node cluster; blue represents positive, red represents negative, and saturation represents magnitude. **(b) Eigenvector Basis** In contrast, eigenvectors are global (low sparsity), distributing mass evenly over the graph. (c) **Haar locality (hop-energy):** For every Haar vector $h_k$, set the per-node energy $e_i(h_k) = h_k(i)^2/\|h_k\|_2^2$. Let $i^\star = \arg\max_i |h_k(i)|$, and for hop shells $s \in \{0, \dots, 4\}$, compute $\sum_{i:\,\mathrm{dist}(i,i^\star)=s} e_i(h_k)$. The heatmap shows that more than $85\%$ of the energy is inside two hops, which shows that it is tightly confined within one hop neighborhood. (d) **Eigenvector Basis:** Eigenvectors distribute energy more uniformly across hops 0–4, signifying inadequate localization and demonstrating the merging of distant cluster signals (hub aliasing). For comparisons of polynomial bases, see App. F.1.

We then filter the node features of every hierarchy graph as, $H^{(\ell)} = U^{(\ell)} \Lambda^{(\ell)} U^{(\ell)\top} X^{(\ell)}$, where $\Lambda^{(\ell)}$ is a learnable diagonal matrix of low- and high-pass gains. This ensures full coverage of every node and cluster with no redundancy. This operation is diagonal in the Haar basis (no cross-mixing). The low-frequency gain $g_{\mathrm{sc}}^{(\ell)}$ and wavelet gains $\gamma_j^{(\ell)}$ are learned.

### 4.3 PREDICTION TASK

For node-level tasks, we compute each node's final embedding by additive unpooling $\widehat{H}_j = H_j^{(0)} + \sum_{\ell=1}^{L} \sum_{i:j\in C_i^{(\ell)}} H_i^{(\ell)}$, so that every node carries its base feature plus the pooled summaries of all clusters across different scales it belongs to. For graph-level tasks, we coarsen until the number of nodes is $|V^{(L)}| = 1$. We then use the embedding of this final supernode for graph classification. The total loss is defined as $L_{\mathrm{total}} = L_{\mathrm{CE}} + \lambda_{\mathrm{div}} L_{\mathrm{div}}$, where $L_{\mathrm{CE}}$ is the standard cross-entropy,

$$L_{\mathrm{div}} = -\sum_{\ell=0}^{L-1} \frac{1}{|V^{(\ell)}|} \sum_{i=1}^{|V^{(\ell)}|} \sum_{k=1}^{K_\ell} A_{s,ik}^{(\ell)} \log A_{s,ik}^{(\ell)},$$

maximizes the entropy of each node's $(A_s^{(\ell)})_{i:}$. $\lambda_{\mathrm{div}}$ balance the auxiliary terms.

### 4.4 THEORETICAL PERSPECTIVE

Because $U[W_{scaling}, W_{inter}, W_{intra}] \in \mathbb{R}^{N \times N}$ is orthonormal, projecting any signal onto these subspaces separates its energy into *global average*, *inter-cluster contrast*, and *intra-cluster variation* without leakage. The basis has both low-frequency and high-frequency directions. The scaling wavelets $W_{scaling}$ capture degree-weighted low-frequency directions of the cluster. The inter-cluster wavelets $W_{inter}$ capture the differences (high frequency content) across neighbouring clusters by separating differences at the boundaries. In Fig. 3, we showed how Haar basis energy is only found within a two-hop, closely linked neighbor. Spectral convolution employing the Haar basis enhances or preserves high-frequency signals for small spokes without expanding or mixing signals with the rest of the graph nodes or cluster, thereby preventing *hub domination* and *oversmoothing*. Finally, the hierarchical coarsening shortens communication pathways (logarithmically in depth), which keeps Jacobian norms from getting too close to zero and prevents *oversquashing*. The subsequent theorems formalize the aforementioned claims.

**Theorem 3** *A spectral filter based on orthonormal and local basis holds any region-specific signal pattern confined under filtering. (proof in Appendix A.3)*

**Theorem 4** *Let $A, B, H$ post-filter centroids are $\mu_A, \mu_B, \mu_H$ and set $\Delta_{AB} = \|\mu_A - \mu_B\|, \Delta_{AH} = \|\mu_A - \mu_{H'}\|$, then $\frac{\Delta_{AB}}{\Delta_{AH}} \geq \sqrt{\frac{M}{a+b}}\left(1 - \frac{2}{\sqrt{M}}\right) > 1$. Thus HSL model avoids hub domination regardless of the hub size $M$. ( proof is in Appendix A.4)*

**Theorem 5** *Regardless of the number of layers in the proposed HMH layer, it overcomes the over-smoothing and oversquashing problem. (Proof is in Appendix A.5)*

Table 1: Graph-classification accuracy (%) comparison on TU datasets. Mean is $\pm 95\%$ CI and '-' means no result found

| Method | PROTEINS | NCI1 | NCI109 | MUTAG | D&D | IMDB-M | REDDIT-12K | Mutagenicity |
|---|---|---|---|---|---|---|---|---|
| GCN | 75.17±3.63 | 76.29±1.79 | 75.91±1.84 | 69.50±1.78 | 73.26±4.46 | 50.39±0.41 | 44.3±1.6 | 79.81±1.58 |
| GraphSAGE | 74.01±4.27 | 74.73±1.34 | 74.17±2.89 | 71.39±1.53 | 75.78±3.91 | 48.13±1.36 | 45.3±0.6 | 78.75±1.18 |
| GAT | 74.72±4.01 | 74.90±1.72 | 75.81±2.68 | 70.81±1.68 | 77.30±3.68 | 45.67±2.70 | 43.9±1.8 | 77.89±2.05 |
| DGCNN | 79.99±0.44 | 74.08±2.19 | 78.23±1.31 | 76.3±1.6 | 70.06±1.21 | 81.34±2.68 | – | 80.41±1.02 |
| DiffPool | 68.90±2.95 | 77.73±0.83 | 77.13±1.49 | 79.22±1.02 | 78.61±1.32 | 51.31±0.72 | 44.8±1.5 | 80.78±1.12 |
| EigenPool | 70.84±1.06 | 77.24±0.96 | 75.99±1.42 | – | 78.63±1.36 | 49.81±0.48 | 44.23±1.3 | 80.11±0.73 |
| gPool | 71.71±1.75 | 76.25±1.39 | 76.61±1.39 | 67.85±1.38 | 77.02±1.32 | 48.3±1.6 | 44.35±0.76 | 80.30±1.54 |
| SAGPool(G) | 71.72±2.19 | 77.88±1.59 | 75.74±1.47 | 76.78±2.12 | 78.70±2.29 | 49.47±0.56 | 42.3±1.6 | 79.72±0.79 |
| GMT | 75.09±0.59 | 76.35±2.62 | – | 84.44±1.33 | 78.72±0.59 | 50.66±0.82 | 43.63±2.6 | 81.32±2.32 |
| TopKPool | 70.48±1.01 | 67.61±3.36 | 73.63±0.55 | 77.61±3.36 | 73.63±0.55 | 48.59±0.72 | **45.3±1.43** | 82.45±1.32 |
| SEP (Wu et al., 2022) | 75.22±0.63 | 74.83±1.49 | 76.58±1.04 | 85.83±1.49 | 78.58±1.04 | 50.78±0.75 | 45.3±1.6 | 79.1±1.2 |
| HMH (ours) | **75.8±2.1** | **78.9±2.5** | **80.7±2.0** | **94.50±1.8** | **79.7±1.8** | **52.5±2.2** | 43.1±1.2 | **81.7±1.8** |

## 5 EXPERIMENT AND RESULT

**Setup.** We evaluate HSL on standard *graph-classification* and *node-classification* benchmarks. All results are reported as mean $\pm$ 95% confidence intervals over 10 random splits. Baselines are run from authors' public implementations with hyperparameters tuned on validation sets. The details of the complete protocol are presented in Appendix A.7

**Synthetic Datatsets**: In addition to the theoretical guarantee provided in theorem 4, we construct a synthetic dataset and measure the cluster separation ratio $r(L)$ for $L = 0, \ldots, 10$ message propagation layers. As shown in Figure 4, baseline models exhibit rapid (approximately exponential) decay in $r(L)$, indicating collapse, whereas HSL preserves the separability between clusters $A$ and $B$ even in the presence of a large hub $H'$.

**Graph Classification.** We evaluate on TU graph-classification benchmarks: PROTEINS, NCI1, NCI109, Mutagenicity, IMDB, and REDDIT-12K. Dataset characteristics and hyperparameter choices are provided in Appendix A.6. We choose well-regarded Hierarchical pooling and strong graph-classification models as the baseline, such as DiffPool (Ying et al., 2018), SAGPool (Lee et al., 2019), TopKPool (Diehl, 2019), Graph U-Net (Gao & Ji, 2019), ASAP (Ranjan et al., 2020), and graph classifiers DGCNN (Phan et al., 2018), SEP (Wu et al., 2022)

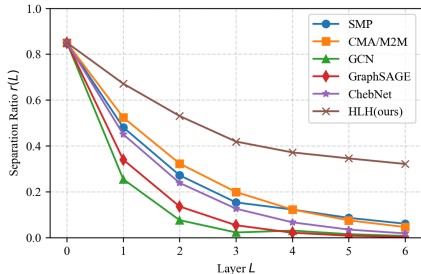

Figure 4: Hub separation ratio

**Graph Classification Results.** HSL yields consistent gains over all baselines (Table 1), improving test accuracy by **+7%** on **MUTAG** and **+2%** on **PROTEINS**. On molecular benchmarks (EN-ZYMES, PROTEINS), our method matches or exceeds diffusion-based pooling approaches (Diff-Pool, EigenPool) while requiring no dense, expensive eigendecomposition. Ablations study on the diffusion barrier, basis components, and gain parameterization are provided in Appendix 6

**Node Classification.** Various benchmark datasets spanning homophilous (Cora, Citeseer, Pubmed, *etc.*) and heterophilous (Chameleon, Squirrel, Texas *etc.*) graphs are used for performing the node classification task. In Appendix B, the attributes of the datasets, Preprocessing, and hyperparameter settings are presented. As suggested by Platonov et al. (2023), we use the *filtered* variants (with duplicate edges removed) for the Chhamelon and Squirrel datasets.

**Node Classification Result.** We choose well-regarded, leading spectral/polynomial-filter methods as Baseline, such as SIGN (Frasca et al., 2020), GPR-GNN (Chien et al., 2020), EvenNet (Lei et al.,

Table 2: Node Classification accuracy (%) comparison with baseline models across all datasets.

| Method | Cora | Citeseer | Actor | Chameleon | Squirrel | Texas | Wisconsin | Cornell |
|---|---|---|---|---|---|---|---|---|
| FAGCN | $87.42 \pm 2.10$ | $76.35 \pm 1.70$ | $35.67 \pm 0.90$ | $36.98 \pm 2.30$ | $42.20 \pm 1.80$ | $77.00 \pm 7.70$ | $86.55 \pm 2.36$ | $84.41 \pm 3.80$ |
| ACM–GCN | $87.91 \pm 0.90$ | $77.32 \pm 1.70$ | $36.28 \pm 1.00$ | $38.93 \pm 1.80$ | $44.40 \pm 1.80$ | $87.84 \pm 4.40$ | $74.19 \pm 3.15$ | $88.14 \pm 6.00$ |
| ASGC | $85.35 \pm 0.98$ | $76.52 \pm 0.36$ | $35.41 \pm 0.80$ | $37.38 \pm 1.06$ | $46.91 \pm 0.89$ | $75.34 \pm 2.30$ | $87.25 \pm 1.63$ | $87.32 \pm 2.30$ |
| GPR–GNN | $82.82 \pm 1.30$ | $70.28 \pm 1.40$ | $31.47 \pm 1.60$ | $39.27 \pm 2.30$ | $46.09 \pm 1.30$ | $87.91 \pm 1.32$ | $91.08 \pm 1.79$ | $90.57 \pm 1.96$ |
| ChebNet | $79.72 \pm 1.10$ | $70.48 \pm 1.00$ | $27.42 \pm 2.30$ | $41.20 \pm 1.20$ | $39.82 \pm 0.80$ | $86.28 \pm 2.62$ | $91.71 \pm 1.62$ | $83.91 \pm 2.17$ |
| ChebNet II | $83.95 \pm 0.80$ | $71.76 \pm 1.20$ | $33.48 \pm 1.20$ | $42.50 \pm 3.10$ | $46.40 \pm 1.10$ | $88.28 \pm 1.47$ | $90.45 \pm 1.22$ | $88.30 \pm 1.48$ |
| BernNet | $82.96 \pm 1.10$ | $73.25 \pm 1.00$ | $37.92 \pm 0.80$ | $39.80 \pm 3.40$ | $44.68 \pm 1.50$ | $86.62 \pm 1.37$ | $91.72 \pm 1.27$ | $92.13 \pm 1.64$ |
| JacobiConv | $84.12 \pm 0.70$ | $72.59 \pm 1.40$ | $36.61 \pm 0.70$ | $\mathbf{42.90 \pm 1.90}$ | $48.65 \pm 0.80$ | $\mathbf{93.44 \pm 2.13}$ | $92.98 \pm 1.84$ | $92.95 \pm 2.46$ |
| OptBasisGNN | $81.97 \pm 1.20$ | $76.46 \pm 1.60$ | $38.84 \pm 1.30$ | $39.70 \pm 2.40$ | $45.66 \pm 1.10$ | $87.32 \pm 0.80$ | $91.55 \pm 1.68$ | $89.43 \pm 4.50$ |
| Specformer | $82.27 \pm 0.70$ | $73.45 \pm 1.40$ | $40.12 \pm 0.60$ | $41.10 \pm 1.20$ | $48.24 \pm 0.90$ | $91.34 \pm 3.40$ | $91.33 \pm 1.95$ | $\mathbf{93.43 \pm 3.50}$ |
| **HSL (ours)** | $\mathbf{88.9 \pm 0.6}$ | $\mathbf{79.2 \pm 0.8}$ | $\mathbf{43.3 \pm 0.7}$ | $41.9 \pm 1.2$ | $\mathbf{50.9 \pm 0.9}$ | $91.5 \pm 1.0$ | $\mathbf{92.26 \pm 1.85}$ | $\mathbf{93.7 \pm 0.9}$ |

2022), ChebNet II (He et al., 2022), BernNet (He et al., 2021), JacobiConv (Wang & Zhang, 2022), and UniFilter (Huang et al., 2024). Across most of the dataset, our model consistently outperforms (Table 2) all baselines with gains up to 2% on citation networks and up to 3% on highly heterophilous graphs squirrel, demonstrating that multi-scale Haar filtering yields more discriminative embeddings than fixed, local polynomial spectral filters.

**Ablation Study.** We ablate four components of **HSL**. (i) **Topology-only mixing:** Excluding $L_{\text{feat}}$ from equation (4) reduces the accuracy on feature-informative datasets, such as Mutagenicity by $\sim 2.1\%$ and Chameleon by $\sim 2.3\ \%$. (ii) **No diffusion barrier:** By Substituting ($\lambda = 0$ in equation (6) we drop $M_C$ constraints, which then allows unrelated nodes to cluster together reducing the accuraccy of Chameleon by $\sim 5.2\ \%$ and Squirrel by $\sim 4.8\ \%$ (iii) **No degree-aware basis:** Substituting a degree-aware basis of equation (14) by a eigendecomposition of $L_{mix}$ incurs additional $(n^2)$ computation cost and reduces the accuracy of the graph learning task; reducing accuracy for Actor by $\sim 1.6\,$pp and Squirrel $\sim 1.9\,$pp (v) **Without balanced relaxation:** We omit projection $U s_c^{(0)} = 0$ in equation (7) during subspace iteration, which may allow the eigenvector to carry a constant/near-constant direction impairing cluster balance. The accuracy of a highly heterogeneous graph, such as Chameleon, dropped by $\sim 6\ \%$.

**Scalability.** HSL scales linearly with graph size. For $n = |V|$ nodes, $m = |E|$ edges, and feature dimension $d$, each forward/backward pass costs $\mathcal{O}(md + nd)$. The memory cost is $\mathcal{O}(n + m)$, storing only sparse adjacencies and wavelets. In contrast, almost every graph pooling baselines incur $\mathcal{O}(n^2)$ time complexity Li et al. (2024).

Table 3: Ablation study on **HSL** variants (mean±std over 5 seeds).

| Variant | Mutagenicity | Actor | Chameleon | Squirrel |
|---|---|---|---|---|
| Full HSL (ours) | $\mathbf{94.4 \pm 4.8}$ | $\mathbf{43.3 \pm 0.7}$ | $\mathbf{41.8 \pm 1.2}$ | $\mathbf{50.9 \pm 1.4}$ |
| Topology-only | $89.8 \pm 0.7$ | $41.1 \pm 1.0$ | $39.5 \pm 1.5$ | $49.2 \pm 1.4$ |
| w/o barrier $M_C$ | $88.1 \pm 0.6$ | $38.9 \pm 1.2$ | $36.6 \pm 1.7$ | $46.1 \pm 1.6$ |
| w/o degree-aware Haar | $89.0 \pm 0.6$ | $41.7 \pm 0.9$ | $40.1 \pm 1.3$ | $49.0 \pm 1.5$ |
| w/o balanced relaxation | $88.4 \pm 0.5$ | $37.8 \pm 1.3$ | $37.4 \pm 1.8$ | $45.0 \pm 1.7$ |

## 6 CONCLUSION

We introduced HSL, a locality-first, multi-view, multiresolution framework for learning on heterophilous graphs. HSL estimates diffusion neighborhoods and enforces a diffusion barrier to prevent spurious co-clustering, then solves a balanced spectral relaxation to obtain soft assignments for hierarchical coarsening. At each level, it builds a strictly local, degree-aware, *orthonormal* Haar basis—whose energy is confined to small neighborhoods—and applies diagonal spectral filtering with learned gains, amplifying boundary-aligned (high-frequency) contrasts while smoothing homogeneous (low-frequency) regions. This design avoids dense global mixing and expensive full eigendecompositions, operates in near-linear time, and directly mitigates hub dominations and oversquashing. Theoretically, we establish the effectiveness of multiresolution learning with respect to the basis that can adapt and respect the locality of graph regions. Empirically, HSL matches prior state-of-the-art on node classification and delivers strong improvements on graph classification, while retaining linear scalability. Future research will extend HSL to dynamic graphs and temporal graphs.

## REPRODUCIBILITY STATEMENT

The algorithm is comprehensively described in our Methodology section, enabling readers to easily understand and reimplement it. We've also uploaded our entire code as supplemental information. Each section of the code has been clearly marked to correspond to the related theory in the Methodology, allowing for easy cross-referencing. In the Appendix, we include a table of all hyperparameters used in the study. The supplementary code includes a Jupyter notebook for graph classification, where we train and test our model on the MUTAG datasets, demonstrating that it achieves 95% test accuracy and outperforms a well-known state-of-the-art method by at least 7 percentage points. We provide a Jupyter notebook for node classification on the heterophilous Cornell dataset, in which we train and test both our approach and the strong baseline GPRGNN, and present a side-by-side comparison of the findings.

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

# A  APPENDIX

## A.1  PROOF OF THEOREM 1

**Case I: Hub Domination in SMP:** Consider three clusters $A$, $B$, and $H'$. The tiny clusters or spokes $A$ and $B$ of size $a = |A|$ and $b = |B|$ are attached to a common hub $H'$ of size $M = |H'|$ with $M \gg a, b$. Let the cluster mean features for $A$, $B$, and $H'$ is $\mu_A \in \mathbb{R}^d$, $\mu_B \in \mathbb{R}^d$, $\mu_{H'} \in \mathbb{R}^d$, respectively. To proceed further, we need the following two assumptions:

**Assumption** (i) The spoke–spoke expected representation gap is small compared to the spoke–hub expected representation gap, reflecting that the hub's expected representation is well separated from that of the spokes. We can define the base spoke–spoke and spoke–hub expected gaps as

$$\delta_0 = \mathbb{E}_{u \in A, \, v \in B} \left\| h_u^{(0)} - h_v^{(0)} \right\|_2, \qquad \Delta_0 = \mathbb{E}_{u \in A, \, h \in H} \left\| h_u^{(0)} - h_h^{(0)} \right\|_2. \tag{15}$$

So the ultimate spoke-spoke and spoke-hub gaps can be written as

$$\delta = \left\| \mu_A - \mu_B \right\|_2, \text{ and } \Delta = \left\| \mu_u - \mu_h \right\|_2. \tag{16}$$

(ii) The hub's size $M$ is sufficiently larger than the combined size with the weighed geometric factor $(\Delta/\delta)$, i.e., $M \geq \frac{(a+b)\Delta}{\delta}(1+\varepsilon)$ for some $\varepsilon > 0$.

Let the signed adjacency is $S_{uv} = +1$ if $(u, v) \in E$ and $y_u = y_v$, and $S_{uv} = -1$ if $y_u \neq y_v$. The SMP update is given by

$$H^{(k+1)} = S H^{(k)} W^{(k)}, \quad \|W^{(k)}\|_2 = 1, \quad H^{(0)} = X. \tag{17}$$

Taking the norm in equation (17), we can write

$$\left\| S H^{(k)} W^{(k)} \right\|_2 \leq \left\| S H^{(k)} \right\|_2.$$

In addition, two consecutive steps of the SMP can be expressed as

$$H^{(k+2)} = S\left( S H^{(k)} W^{(k)} \right) W^{(k+1)} \approx S^2 H^{(k)}.$$

Since there are no edges between $A$ and $B$, a walk from a node in $A$ to another region of the graph must pass through $H'$. Similarly, a walk from a node in $B$ to another region must pass through $H'$. Hence for $u \in A$, $v \in B$, and any $h \in H'$ the second layer embedding can be expressed as

$$h_u^{(2)} = (S^2 X)_u = \sum_{w \in H} \sum_{z \in N(w) \cap A} X_z$$
$$+ \sum_{z \in N(u) \cap A} \sum_{w \in N(z) \cap H} X_w \approx M \mu_H + (a-1)\mu_A, \tag{18}$$

Similarly, for node $v$ and $h$ the following can be written as follows,

$$h_v^{(2)} \approx M \mu_H' + (b-1)\mu_B, \tag{19}$$

$$h_h^{(2)} \approx M \mu'_H + (a-1) \mu_A + (b-1) \mu_B. \tag{20}$$

By subtracting equation (18) and equation (19), taking the norm, and applying the definition in equation (16), we obtain

$$\begin{aligned} \|h_u^{(2)} - h_v^{(2)}\|_2 &\leq \|(a-1) \mu_A - (b-1) \mu_B\|_2 \\ &\leq (a+b) \|\mu_A - \mu_B\|_2 = (a+b) \delta. \end{aligned} \tag{21}$$

By subtracting equation (18) and equation (20), taking the norm, and applying the definition in equation (16), we obtain

$$\begin{aligned} \|h_u^{(2)} - h_h^{(2)}\|_2 &\geq \left\| M \mu_H - \left((a-1)\mu_A + (b-1)\mu_B\right) \right\|_2 - (a+b) \delta \\ &\geq M \Delta - (a+b) \delta \end{aligned} \tag{22}$$

Dividing equation (21) by equation (22) gives

$$\frac{\|h_u^{(2)} - h_v^{(2)}\|_2}{\|h_u^{(2)} - h_h^{(2)}\|_2} \leq \frac{(a+b) \delta}{M \Delta - (a+b) \delta} \leq \frac{r}{1+\epsilon}, \quad r = \frac{a+b}{M}. \tag{23}$$

So that after $L = 2k$ layers equation (23) becomes,

$$\frac{\|h_u^{(L)} - h_v^{(L)}\|_2}{\|h_u^{(L)} - h_h^{(L)}\|_2} \leq \left(\frac{r}{1+\epsilon}\right)^k = \left(\frac{r}{1+\epsilon}\right)^{\lceil L/2 \rceil}, \tag{24}$$

Since $r < 1$, we can write $\frac{r}{1+\epsilon} < 1$. Therefore, spoke-spoke separation diminishes exponentially in relation to the spoke-hub separation across the layer of SMP.

**Case II: CMA Setup**: Each node $i$ at layer $\ell$ has embedding $h_i^{(\ell)} \in \mathbb{R}^d$, with fixed base features $h_i^{(0)}$. A linear projection of the embedding can be expressed as

$$z_i := W h_i^{(\ell)} \in \mathbb{R}^{d'}. \tag{25}$$

Before proceeding further, we need the following standard assumptions:
**Assumption 1: Uniform bound.** There exists $F > 0$ such that

$$\|z_j\| \leq F \qquad \text{for all } j \in V. \tag{26}$$

*Remarks.* This is a mild boundedness condition on the projected representations at the working layer $\ell$. If $\|W\|_2 \leq C$ and $\|h_j^{(\ell)}\| \leq B$ satisfy uniformly in $j$, in which case one can write $F = CB$. Thus, the bound is trivially satisfied.
**Assumption 2:** The spoke–spoke expected representation gap is small compared to the spoke–hub expected representation gap, reflecting that the hub's expected representation is well separated from that of the spokes. We can define the base spoke–spoke and spoke–hub expected gaps as

$$\delta_0 = \mathbb{E}_{u \in A, \, v \in B} \left\| h_u^{(0)} - h_v^{(0)} \right\|_2, \text{ and } \Delta_0 = \mathbb{E}_{u \in A, \, h \in H} \left\| h_u^{(0)} - h_h^{(0)} \right\|_2. \tag{27}$$

given the hub dominance, we can assume $\delta_0 < k\Delta_0$ where $0 < k << 1$.
**Assumption 3: Hub dominance.** Let $\pi_{H'}$ is a probability vector based on the hub set $H'$ (for instance, degree-weighted: $\pi_{H'}(h) = \deg(h) / \sum_{h \in H} \deg(h')$). For each node $i \in V$, the attention distribution $\pi_i$ across its message sources decomposes as

$$\pi_i = (1 - \rho_i)\pi_{H'} + \rho_i q_i, \qquad 0 \leq \rho_i \leq \rho < 1, \tag{28}$$

where $q_i$ is an arbitrary probability distribution supported on $N(i)$. A $(1-\rho_i)$ portion of $i$'s attention mass is pooled from the hub via the fixed hub mixture $\pi_{H'}$, while the other $\rho_i$ fraction is shared among its local neighbors according to $q_i$. The global constant $\rho < 1$ sets the maximum value for all $\rho_i$ and measures the "strength of hub dominance"– the smaller $\rho$, the stronger the dominance of hub messages.
The aggregated message is for node $i$ in the graph $G$ is

$$m_i := \sum_j \pi_i(j) z_j. \tag{29}$$

According to the CMA algorithm, the single-layer update is defined as

$$h_i^{(\ell+1)} := (1-\beta)h_i^{(0)} + \beta m_i, \qquad \beta \in (0,1]. \tag{30}$$

We can define the hub means in the projected space by using the probability vector $\pi_{H'}$ as

$$\mu_{H'}^z := \sum_{h \in H'} \pi_{H'}(h)z_h. \tag{31}$$

From equation (28) and equation (29), we can expand the updated mesage $m_i$ for node $i$ as follows,

$$m_i = \sum_j \big[(1-\rho_i)\pi_{H'}(j) + \rho_i q_i(j)\big]z_j = (1-\rho_i)\mu_{H'}^z + \rho_i \tilde{m}_i, \tag{32}$$

where $\tilde{m}_i := \sum_j q_i(j)z_j$.

Since $q_i$ and $\pi_{H'}$ are probability vectors and by equation (26) in the Assumption, we can rewrite following expression

$$\|\tilde{m}_i\| \le \sum_j q_i(j)\|z_j\| \le F, \qquad \|\mu_{H'}^z\| \le \sum_{h \in H} \pi_{H'}(h)\|z_h\| \le F. \tag{33}$$

Using the bounds in equation (33), the equation (32) can be rewritten as follows,

$$m_i = \mu_{H'}^z + \rho_i(\tilde{m}_i - \mu_{H'}^z). \tag{34}$$

For any $u \in A$ and $v \in B$, using equation (34),

$$m_u - m_v = \rho_u(\tilde{m}_u - \mu_{H'}^z) - \rho_v(\tilde{m}_v - \mu_{H'}^z). \tag{35}$$

Taking norms and applying the triangle inequality to 35 we get,

$$\|m_u - m_v\| \le \rho_u\|\tilde{m}_u - \mu_{H'}^z\| + \rho_v\|\tilde{m}_v - \mu_{H'}^z\|. \tag{36}$$

For a node $u \in A$ using equation (33), we can wrtie

$$\|\tilde{m}_u - \mu_{H'}^z\| \le \|\tilde{m}_u\| + \|\mu_{H'}^z\| \le 2F. \tag{37}$$

Similarly, we can prove the following for node $v \in B$:

$$\|\tilde{m}_v - \mu_{H'}^z\| \le \|\tilde{m}_u\| + \|\mu_{H'}^z\| \le 2F \tag{38}$$

Combinining equation (37) and equation (38) and given $\rho_u, \rho_v \le \rho$, equation (36) can be written as

$$\|m_u - m_v\| \le 4\rho F. \tag{L1}$$

Substituting $i = u$ and $i = h$ into equation (34) and substracting them follows,

$$m_u - m_h = \rho_u(\tilde{m}_u - \mu_{H'}^z) - \rho_h(\tilde{m}_h - \mu_{H'}^z).$$

The same bound as in equation (36), equation (37), and equation (38) applies, yields,

$$\|m_u - m_h\| \le 4\rho F. \tag{L2}$$

For $u \in A, v \in B$, using equation (30), we can write

$$h_u^{(\ell+1)} - h_v^{(\ell+1)} = (1-\beta)(h_u^{(0)} - h_v^{(0)}) + \beta(m_u - m_v). \tag{39}$$

Taking norms, using equation (L1), and recalling $\mathbb{E}_{u \in A, v \in B} \|h_u^{(0)} - h_v^{(0)}\| = \delta_0$ the equation (39) can be written as

$$\|h_u^{(1)} - h_v^{(1)}\| \le (1-\beta)\delta_0 + 4\beta\rho F, \tag{40}$$

Similarly for $u \in A$ and $h \in H$ we can write using equation (30)

$$h_u^{(\ell+1)} - h_h^{(\ell+1)} = (1-\beta)(h_u^{(0)} - h_h^{(0)}) + \beta(m_u - m_h). \tag{41}$$

Using reverse triangle inequality, using equation (L2) and recalling $\mathbb{E}_{u \in A, h \in H} \|h_u^{(0)} - h_h^{(0)}\| = \Delta_0$ yields

$$\|h_u^{(1)} - h_h^{(1)}\| \geq (1 - \beta)\Delta_0 - 4\beta\rho F, \tag{42}$$

Now define the per-layer separation ratio

$$r^{(1)} = \frac{\mathbb{E}_{u \in A, v \in B} \|h_u^{(1)} - h_v^{(1)}\|}{\mathbb{E}_{u \in A, h \in H} \|h_u^{(1)} - h_h^{(1)}\|}. \tag{43}$$

Using the bound in equation (40) and equation (42) we can rewrite equation (43) as follows,

$$r^{(\ell+1)} \leq \frac{(1 - \beta)\delta_0 + 4\beta\rho F}{(1 - \beta)\Delta_0 - 4\beta\rho F} = \eta. \tag{R1}$$

The denominator in equation (R1) is $(1 - \beta)\Delta_0 - 4\beta\rho F$. For fixed $\beta \in (0, 1)$ and $F > 0$ it is positive whenever

$$(1 - \beta)\Delta_0 > 4\beta\rho F \quad \iff \quad \rho < \frac{1 - \beta}{4\beta F} \Delta_0.$$

Thus, decreasing $\rho$ (stronger hub dominance) or increasing $\Delta_0$ (clearer hub–spoke margin) ensures the denominator is positive. Employing Assumption 2, we get $\eta < 1$. So the R1 can be writtens as

$$\frac{\|h_u^{(L)} - h_v^{(L)}\|}{\|h_u^{(L)} - h_h^{(L)}\|} \leq \eta \tag{44}$$

where $\eta < 1$. For the L layer, we can write,

$$\frac{\|h_u^{(L)} - h_v^{(L)}\|}{\|h_u^{(L)} - h_h^{(L)}\|} \leq (\eta)^L. \tag{45}$$

We can state that the spoke–spoke separation decays *geometrically* relative to the spoke–hub separation as depth increases, so CMA concentrates information toward the hub.

**Case III: Global Basis.** Let $G = (V, E, w)$ be an undirected weighted graph with a symmetric Laplacian $L \in \mathbb{R}^{n \times n}$. Its eigendecomposition is

$$L = U\Lambda U^\top, \quad U = [u_1 \; \cdots \; u_n], \quad U^\top U = I, \quad \Lambda = \operatorname{diag}(\lambda_1, \ldots, \lambda_n), \tag{46}$$

where $\{u_k\}_{k=1}^n$ are orthonormal eigenvectors with eigenvalues $\{\lambda_k\}_{k=1}^n$. According to the figure 1, there are no edges between $A$ and $B$. We can define the global spectral filtering $f : \mathbb{R} \to \mathbb{R}$ as

$$\Phi = f(L) = U f(\Lambda) U^\top. \tag{47}$$

A *global spectral filter* is any matrix function

$$\Phi = f(L) = U f(\Lambda) U^\top, \qquad f(\Lambda) = \operatorname{diag}(f(\lambda_1), \ldots, f(\lambda_n)).$$

For $i \in V$ let $e_i \in \mathbb{R}^n$ be the coordinate vector and set the $i$th row of $\Phi$ as $r_i^\top := e_i^\top \Phi$.

**Assumption: Hub Dominance.** For each eigenvector $u_k$, we can define the degree-weighted hub mean as

$$\mu_k(H) := \frac{\sum_{h \in H} d_h \, u_k(h)}{\sum_{h \in H} d_h}.$$

Then, there exist nonnegative numbers $\{\delta_k\}_{k=1}^n$ such that

$$\max_{i \in A \cup B} \left| u_k(i) - \mu_k(H) \right| \leq \delta_k \qquad \text{for all } k = 1, \ldots, n. \tag{48}$$

**Remark 1** *Hub dominance ensures that $\delta_k$ is small. At any spoke node $i \in A \cup B$, the eigenvector equation $(Lu_k)(i) = \lambda_k u_k(i)$ forces $u_k(i)$ to be a weighted average of neighbor values. When most of $i$'s neighbors lie in the hub $H$, $u_k(i)$ must be close to the hub average $\mu_k(H)$, resulting in a small $\delta_k$. Larger and denser hubs shrink $\delta_k$ further. This hub-dominance condition informally states that each eigenvector is nearly flat on the spoke sets and tied to the hub average.*

%[ The $i$-th row of $\Phi$ can be written as,

$$r_i^\top = e_i^\top \Phi = e_i^\top U f(\Lambda) U^\top = \sum_{k=1}^{n} f(\lambda_k) u_k(i) u_k^\top. \tag{49}$$

For $a \in A, b \in B$ using the equation (49), we can write as follows,

$$r_a^\top - r_b^\top = \sum_{k=1}^{n} f(\lambda_k) \big( u_k(a) - u_k(b) \big) u_k^\top. \tag{50}$$

Since $\{u_k\}$ forms an orthonormal basis, Parseval's identity Poulin (2020) gives

$$\|r_a^\top - r_b^\top\|_2^2 = \sum_{k=1}^{n} f(\lambda_k)^2 \big( u_k(a) - u_k(b) \big)^2. \tag{51}$$

From the hub-dominance condition equation (48),

$$|u_k(a) - u_k(b)| \leq |u_k(a) - \mu_k(H)| + |u_k(b) - \mu_k(H)| \leq 2\delta_k. \tag{52}$$

Plugging equation (52) into equation (51) and taking square roots yields

$$\|e_a^\top \Phi - e_b^\top \Phi\|_2 \leq 2 \left( \sum_{k=1}^{n} f(\lambda_k)^2 \, \delta_k^2 \right)^{1/2}, \tag{53}$$

where $r_i = e_i \phi$. Aso plugging equation (52) into equation (51) gives

$$\left\| r_a - r_b \right\|_2 \leq 2 \left\| f(\Lambda) \, \boldsymbol{\delta} \right\|_2. \tag{54}$$

Since $(\Phi X W)_i = r_i^\top X W$, we have

$$\|(\Phi X W)_a - (\Phi X W)_b\|_2 = \|(r_a^\top - r_b^\top) X W\|_2 \leq \|r_a^\top - r_b^\top\|_2 \|X W\|_{2 \to 2}, \tag{55}$$

Using equation (54), we can rewrite the equation (55) as

$$\|(\Phi X W)_a - (\Phi X W)_b\|_2 \leq \|X W\|_{2 \to 2} \cdot 2 \left( \sum_{k=1}^{n} f(\lambda_k)^2 \, \delta_k^2 \right)^{1/2}. \tag{56}$$

**Multi-hop message passing.** Take the random-walk Laplacian $L_{\mathrm{rw}} = I - P$ where $P = D^{-1}A$. A pure $k$-hop propagation equals

$$\Phi_k = P^k = (I - L_{\mathrm{rw}})^k = f_k(L_{\mathrm{rw}}), \quad f_k(\lambda) = (1 - \lambda)^k. \tag{57}$$

Now given the k-hop filtering is $f_k$ and $L = L_{\mathrm{rw}}$ gives, for any $a \in A, b \in B$. we obtain

$$\|e_a^\top P^k - e_b^\top P^k\|_2 \leq 2 \left( \sum_{j=1}^{n} (1 - \lambda_j)^{2k} \, \delta_j^2 \right)^{1/2}. \tag{58}$$

For $L_{\mathrm{rw}}$, the trivial eigenpair is $\lambda_1 = 0$ with $u_1 \propto \mathbf{1}$, so $\delta_1 = 0$. Given that the graph is aperiodic, we can define

$$\rho := \max_{j \geq 2} |1 - \lambda_j| < 1. \tag{59}$$

Then. it holds that

$$\|e_a^\top P^k - e_b^\top P^k\|_2 \leq 2\rho^k \left( \sum_{j=2}^{n} \delta_j^2 \right)^{1/2}. \tag{60}$$

Thus, the difference contracts exponentially in hop count $k$. The same reasoning applies to any polynomial-in-$P$ (or $L$) message-passing kernel. So using the fact that $r_{=} e_a P^k$ we can write

$$\|r_a^\top P^k - r_b^\top P^k\|_2 \leq 2\rho^k \left( \sum_{j=2}^{n} \delta_j^2 \right)^{1/2}. \tag{61}$$

Substituting equation (61) into equation (55) gives

$$\|(\Phi XW)_a - (\Phi XW)_b\|_2 \le \|XW\|_2.2\left(\sum_{k=1}^n \rho^k \delta_{(k-1)}^2\right)^{1/2}. \tag{62}$$

Defining

$$C = \|XW\|_2 \cdot 2\left(\sum_{j=1}^n \delta_j^2\right)^{1/2} < 1,$$

We obtain the concise rate bound

$$\|(\Phi XW)_a - (\Phi XW)_b\|_2 \le C \cdot \rho^k.$$

Assume the feature head is normalized, $\|XW\|_{2\to2} \le 1$. Now Given the $\delta_k << 1$ and $0 \le \rho < 1$, the geometric factor $\rho^k$ dominates the decay. So the embeddings contract at the strict geometric rate $\rho \in (0,1)$. Hence, as $k \to \infty$, the right-hand side decays to zero exponentially, and the embeddings of $a$ and $b$ become indistinguishable.

## A.2  THEOREM 2

For $U \in \mathbb{R}^{|C|\times K}$ with $U^\top U = I$ and $U^\top s_C^{(0)} = 0$, define

$$E_\lambda(U) := \mathrm{Tr}\big(U^\top(L_{\mathrm{mix},C} + \lambda M_C)U\big). \tag{63}$$

Since $Y^\star \in \mathbb{R}^{|C|\times K}$, denote its degree-normalized indicator matrix according to diffusion, cohesion, and diffusion margin we can write $\mathrm{Tr}(Y^{\star\top} M_C Y^\star) = 0$. Hence we can write,

$$E_\lambda(Y^\star) = \Phi_{\mathrm{mix}}(S^\star). \tag{64}$$

**(Discrete exactness).** Let $S$ be any other balanced partition that merges node across parts of $S^\star$, and let $Y$ be its degree-normalized indicator. By the mixed-geometry gap and incompatibility mass, we can write, $\Psi(S) := \mathrm{Tr}(Y^\top M_C Y)$,

$$E_\lambda(Y) = \Phi_{\mathrm{mix}}(S) + \lambda\Psi(S) \ge \Phi_{\mathrm{mix}}(S^\star) + \Delta_{\mathrm{mix}} + \lambda\Psi_{\mathrm{min}}.$$

Thus, if

$$\lambda \ge \lambda_\star := \Delta_{\mathrm{mix}}/\Psi_{\mathrm{min}}, \tag{3}$$

then $E_\lambda(Y) \ge E_\lambda(Y^\star)$ with strict inequality whenever $S$ merges across parts. Hence, $Y^\star$ is the unique minimizer among balanced indicator matrices.

**(Spectral relaxation).** Consider any feasible $U$ that is orthonormal and balanced. Then, we have
(a) If $\mathrm{Tr}(U^\top M_C U) > 0$, then $U$ mixes across ground truth nodes. Given the diffusion margin

$$\mathrm{Tr}(U^\top M_C U) \ge \Psi_{\mathrm{min}}, \tag{4}$$

using equation (63), equation (64), and equation (A.2), we get

$$E_\lambda(U) \ge \lambda\Psi_{\mathrm{min}} \ge \Phi_{\mathrm{mix}}(S^\star) + \Delta_{\mathrm{mix}} > E_\lambda(Y^\star)$$

Hence no minimizer can have positive barrier.

(b) Let $U^\star$ minimize $E_\lambda$. Since $M_C$ is strictly positive on cross–part pairs (i.e., $(M_C)_{ij} > 0$ whenever $i \in S_k$, $j \in S_\ell$, $k \ne \ell$), the only way for the cross–part node mass $\langle M_C, P^\star\rangle$ to be zero is that $P^\star := U^\star(U^\star)^\top$ has no off-block entries. Equivalently, $P^\star$ is block–diagonal with respect to the ground-truth partition $\mathcal{S}^\star$. So from (a), we can write $\mathrm{Tr}(U^{\star\top} M_C U^\star) = 0$. So given $(M_C)_{ij} > 0$ for all $i \in S_k^\star$, $j \in S_\ell^\star$, $k \ne \ell$ (diffusion margin), $\mathrm{Tr}(U^{\star\top} M_C U^\star) = 0$ forces

$$\big(U^\star U^{\star\top}\big)_{ij} = 0 \quad \text{for all } i \in S_k^\star, \ j \in S_\ell^\star, \ k \ne \ell, \tag{S6}$$

i.e., the projector $U^\star U^{\star\top}$ is block–diagonal w.r.t. $S^\star$. After an orthogonal change of basis in the columns, each column of $U^\star$ can be taken to be supported on a single part $S_k^\star$.

**(c) Within–part optimal directions.** Inside any fixed part $S_k^\star$, the mixed Laplacian Rayleigh quotient is minimized (uniquely, under the balance constraint) by the degree-normalized constant on that part (the corresponding column of $Y^\star$). Any vector orthogonal to that constant has strictly larger Rayleigh energy. So let's fix a part $S_k^\star$. Over vectors $v$ supported on $S_k^\star$ and orthogonal to $s_C^{(0)}$, the Rayleigh quotient

$$R(v) := \frac{v^\top L_{\mathrm{mix},C}\, v}{\|v\|_2^2}$$

is minimized uniquely by the degree–normalized constant $y_k^\star$ on $S_k^\star$. Because we must pick exactly $K$ orthonormal directions overall and each lives in a distinct part by (b), the optimal choice is one constant per part. So given any $v \perp y_k^\star$ has $R(v) > R(y_k^\star)$. Therefore, among all $K$–column block–supported $U$, the minimal trace

$$\mathrm{Tr}\big(U^\top L_{\mathrm{mix},C} U\big) = \sum_{m=1}^{K} R(u_m)$$

is attained by choosing exactly one constant per part, so

$$\mathrm{span}(U^\star) = \mathrm{span}(Y^\star). \tag{65}$$

**(d) Bottom–$K$ eigenspace.** Let $A_\lambda := L_{\mathrm{mix},C} + \lambda M_C$. Any orthonormal basis of $\mathrm{span}(Y^\star)$ minimizes the relaxed objective, hence the optimal relaxed subspace equals the bottom–$K$ eigenspace of $A_\lambda$ on the balanced subspace. If $\lambda_{K+1}(A_\lambda) - \lambda_K(A_\lambda) > 0$, this subspace is unique up to rotation, and standard balanced rounding (e.g., $K$–means on rows or a linear/MLP head) recovers $S^\star$. *Conclusion.* For $\lambda \geq \lambda_\star$, every minimizer of the spectral relaxation spans the degree-normalized indicator subspace of $S^\star$. Equivalently, the bottom-$K$ eigenspace of $L_{\mathrm{mix},C} + \lambda M_C$ equals $\mathrm{span}(Y^\star)$, so balanced rounding (or a simple MLP head) recovers $S^\star$ exactly.

## A.3 PROOF OF THEOREM 3

Set up two distinct, non-overlapping small clusters $A, B \subset V$, next to a significant hub region $H' \subset V$, ensuring that $A \cap B = \varnothing$, $A \cap H' = \varnothing$, $B \cap H' = \varnothing$. Let $x_A$ and $x_B$ be unit-norm normalized indicator signal vectors that exist only in clusters $A$ and $B$. This means that $(x_A)_i = 0$ for every node $i \notin A$.

Let $\mathcal{F} : \mathbb{R}^{|V|} \to \mathbb{R}^{|V|}$ be a single linear filter. A stack of $L$ layers of a filter is defined as $\mathcal{F}^{(L)} = \mathcal{F}_L \cdots \mathcal{F}_1$, where each $\mathcal{F}_\ell$ is a linear filter. The separation ratio of two cluster signals after $L$ filtering layers for pre-filtering embeddings is given by

$$r(L) = \frac{\big\| \mathcal{F}^{(L)} x_A - \mathcal{F}^{(L)} x_B \big\|_2}{\| x_A - x_B \|_2}.$$

The filter used on region A takes the form $\mathcal{F}^{(L)} x_A$, and the filter used on region B is referred to as $\mathcal{F}^{(L)} x_B$. The filter $F^{(L)}$ is diagonal in the localized orthonormal basis, which means that it shifts the scale of each coordinate by the gains $g_k$. We will show that $\langle F^{(L)} x_A, F^{(L)} x_B \rangle = 0$, which means that there is no cross-region leakage. We will also show that the difference between the two signals after filtering stays the same within a scaling factor, precisely, $g_{\min} \leq r(L) \leq g_{\max}$. This suggests that the filter restricts regional patterns and preserves their contrast, modifying only their magnitude through diagonal gains.

Let $T = [t_1, \ldots, t_n] \in \mathbb{R}^{n \times n}$ be an *orthonormal, spatially localized* basis ($T^\top T = TT^\top = I$) whose bases have disjoint supports across the regions $A, B, H$, where every column $t_k$ is supported fully inside exactly one of $A$, $B$, or $H$. We can define the index sets as ,

$$\Omega_A := \{k : \mathrm{supp}(t_k) \subseteq A\}, \quad \Omega_B := \{k : \mathrm{supp}(t_k) \subseteq B\}, \quad \Omega_H := \{1, \ldots, n\} \setminus (\Omega_A \cup \Omega_B).$$

Assume each layer $F_\ell$ ($\ell = 1, \ldots, L$) is diagonal in the $T$-coordinates:

$$F_\ell = T \mathrm{diag}\big(h^{(\ell)}\big) T^\top, \qquad h^{(\ell)} \in \mathbb{R}^n.$$

Then the $L$-layer stack equals

$$F^{(L)} := F_L \cdots F_1 = T \operatorname{diag}(g) T^\top, \qquad g_k := \prod_{\ell=1}^{L} h_k^{(\ell)}.$$

Now, any signal $x \in \mathbb{R}^{|V|}$ can be expanded as $x = Tc$, where $c = T^\top x$. Localization means that an $A$-supported component has coefficients supported only on $\Omega_A$. So we can write,

$$x_A = Tc_A, \quad (c_A)_k = 0 \text{ for } k \notin \Omega_A, \qquad x_B = Tc_B, \quad (c_B)_k = 0 \text{ for } k \notin \Omega_B.$$

Since $T$ is orthonormal, $\langle x_A, x_B \rangle = c_A^\top c_B = 0$, and $\|x\|_2 = \|T^\top x\|_2$.

**No cross-region leakage.** Using $F^{(L)} = T\operatorname{diag}(g)T^\top$,

$$\langle F^{(L)}x_A, \ F^{(L)}x_B \rangle = \big(\operatorname{diag}(g)c_A\big)^\top \big(\operatorname{diag}(g)c_B\big) = \sum_k g_k^2 \, (c_A)_k (c_B)_k = 0,$$

because the supports of $c_A$ and $c_B$ are disjoint.

**Separation bounds.** From orthogonality and disjoint supports,

$$\|x_A - x_B\|_2^2 = \|c_A\|_2^2 + \|c_B\|_2^2. \tag{66}$$

Applying $F^{(L)}$ gives

$$\|F^{(L)}x_A - F^{(L)}x_B\|_2^2 = \|\operatorname{diag}(g)c_A\|_2^2 + \|\operatorname{diag}(g)c_B\|_2^2 = \sum_k g_k^2\big((c_A)_k^2 + (c_B)_k^2\big).$$

Let $g_{\min} = \min_k |g_k|$ and $g_{\max} = \max_k |g_k|$. Then

$$g_{\min}^2\big(\|c_A\|_2^2 + \|c_B\|_2^2\big) \ \leq \ \|F^{(L)}x_A - F^{(L)}x_B\|_2^2 \ \leq \ g_{\max}^2\big(\|c_A\|_2^2 + \|c_B\|_2^2\big).$$

Combining with equation (66) and taking square roots yields the separation ratio

$$r(L) := \frac{\|F^{(L)}x_A - F^{(L)}x_B\|_2}{\|x_A - x_B\|_2} \quad \text{satisfies} \quad g_{\min} \ \leq \ r(L) \ \leq \ g_{\max}.$$

Under diagonal, region-localized filtering, $A$ and $B$ maintain orthogonality regardless of how many layers there are. The only thing that affects their separation is the coordinate-wise gains $\{g_k\}$: the ratio $r(L)$ is squeezed between the smallest and largest magnitudes of those gains.

A.4  PROOF OF THEOREM 4

At Level 2 of the hierarchy, we are applying the HSL algorithm. Let's denote the clusters as A, B, and H, where $A = |a|$, $B = |b|$, $H = |M|$, and $a, b << M$. Now we assume their means by $\mu_{H'}, \mu_A, \mu_B$ with $\mu_A \neq \mu_B$. Also, since we assumed the encoder output $X$ is constant within each cluster, its inner product with any such wavelet (inter-class wavelets) vanishes due to the orthogonal property.

Define the global scaling and cluster-contrast basis vectors as

$$s := \tfrac{1}{M+a+b}\,\mathbf{1},$$

$$w_{A,H} := \sqrt{\tfrac{a(M+a)}{M}}\,\mathbf{1}_A \ - \ \sqrt{\tfrac{M(M+a)}{a}}\,\mathbf{1}_H, \tag{67}$$

$$w_{B,H} := \sqrt{\tfrac{b(M+b)}{M}}\,\mathbf{1}_B \ - \ \sqrt{\tfrac{M(M+b)}{b}}\,\mathbf{1}_H.$$

where $\mathbf{1}_A$, $\mathbf{1}_B$ and $\mathbf{1}_H$ are indicator vectors for clusters $A$ $B$, and $H$ respectively. These are the only basis vectors with nonzero projection on constant-per-cluster signals.

Now define the projector of this basis as

$$P := ss^\top, \qquad W := w_{A,H}w_{A,H}^\top + w_{B,H}w_{B,H}^\top.$$

The Haar filter is defined as

$$\Phi := \lambda_{\mathrm{sc}}P + \lambda_{\mathrm{wav}}W, \qquad 0 < \lambda_{\mathrm{sc}} \leq \lambda_{\mathrm{wav}}. \tag{68}$$

According to the encoder design, it follows that $P + W = I$ on that subspace; every such vector $x$ decomposes uniquely as $x = Px + Wx$, where $Px$ is the hub-dominated mean component and $Wx$ encodes the spoke–hub contrasts. The filter then suppresses the former and amplifies the latter as,

$$\Phi x = \lambda_{\text{sc}} P x + \lambda_{\text{wav}} W x, \qquad \frac{\lambda_{\text{wav}}}{\lambda_{\text{sc}}} = \lambda_{\text{gain}} \gg 1.$$

Let's denote the post-filter value of the node is $h$. For any cluster $T \in \{A, B, H\}$, denote $\bar{h}_T = \frac{1}{|T|} \sum_{i \in T} h_i$ as the mean of embedding $h$ over $T$. The mean embeddings for $A, B$ and $H$ can be defined as

$$\bar{h}_A = \lambda_{\text{sc}} \bar{x} + \lambda_{\text{wav}} \frac{c_A}{a},$$

$$\bar{h}_B = \lambda_{\text{sc}} \bar{x} + \lambda_{\text{wav}} \frac{c_B}{b}, \tag{69}$$

$$\bar{h}_H = \lambda_{\text{sc}} \bar{x} - \lambda_{\text{wav}} \left( \frac{M}{a} c_A + \frac{M}{b} c_B \right).$$

where $\bar{x} = \frac{M \mu_{H'} + a \mu_A + b \mu_B}{M + a + b}$ and $c_A = \frac{aM}{M+a}(\mu_A - \mu_{H'})$, $\qquad c_B = \frac{bM}{M+b}(\mu_B - \mu_{H'})$.

To assess the separation between clusters $A$ and $B$ after filtering, subtract their means and square the means using equation (69), we get

$$\|\bar{h}_A - \bar{h}_B\|^2 = \lambda_{\text{wav}}^2 \frac{M}{a+b} \|\mu_A - \mu_B\|^2. \tag{70}$$

Similarly, for the difference between cluster $A$ and the hub $H$ using equation (69), we get

$$\bar{h}_A - \bar{h}_H = \left( \lambda_{\text{sc}} \bar{x} + \lambda_{\text{wav}} \frac{c_A}{a} \right) - \left( \lambda_{\text{sc}} \bar{x} - \lambda_{\text{wav}} \frac{c_A + c_B}{M} \right)$$

$$= \lambda_{\text{wav}} \left( \frac{c_A}{a} + \frac{c_A + c_B}{M} \right). \tag{71}$$

Then, we have

$$\|\bar{h}_A - \bar{h}_H\|^2 \lesssim \lambda_{\text{wav}}^2 \frac{a}{M} \|\mu_A - \mu_{H'}\|^2. \tag{72}$$

Let's define the two key distances after filtering as

$$\Delta_{AB} := \|\bar{h}_A - \bar{h}_B\|, \quad \Delta_{AH} := \|\bar{h}_A - \bar{h}_H\|, \tag{73}$$

where $\bar{h}_A$, $\bar{h}_B$, and $\bar{h}_H$ are the cluster means of the filtered features on $A$, $B$, and $H$ respectively.

Substituting equation (73) into equation (70) we get

$$\Delta_{AB}^2 = \lambda_{\text{wav}}^2 \frac{M}{a+b} \|\mu_A - \mu_B\|^2, \tag{74}$$

and again substituting equation (73) into equation (72), we have

$$\Delta_{AH}^2 \leq \lambda_{\text{wav}}^2 \frac{a}{M} \|\mu_A - \mu_{H'}\|^2. \tag{75}$$

Dividing equation (74) by equation (75), we obtain

$$\frac{\Delta_{AB}}{\Delta_{AH}} \gtrsim \frac{\sqrt{\lambda_{\text{wav}}^2 \frac{M}{a+b} \|\mu_A - \mu_B\|^2}}{\sqrt{\lambda_{\text{wav}}^2 \frac{a}{M} \|\mu_A - \mu_{H'}\|^2}} \tag{76}$$

$$= \frac{\sqrt{M/(a+b)}}{\sqrt{a/M}} \cdot \frac{\|\mu_A - \mu_B\|}{\|\mu_A - \mu_{H'}\|} \tag{77}$$

$$= \frac{M}{a+b} \cdot \frac{\|\mu_A - \mu_B\|}{\|\mu_A - \mu_{H'}\|}. \tag{78}$$

Now, before applying the basis, we applied the heterophyllous encoder, which gave us the mean embeddings of the cluster.

Work in the $k$-dimensional spectral embedding $X$ whose rows are the top-$k$ eigenvectors (optionally row-normalized). In the ideal block-constant case

$$x_i = \begin{cases} \mu_A & (i \in A), \\ \mu_B & (i \in B), \\ \mu_{H'} & (h \in H), \end{cases}$$

where $\mu_A, \mu_B, \mu_{H'} \in \mathbb{R}^k$ are the spoke and hub means. Normalize the cluster means to have equal norm (standard in spectral clustering; if not, apply layer normalization):

$$\|\mu_A\| = \|\mu_B\| = \|\mu_{H'}\| = 1.$$

**Spoke–spoke margin.** Assume a spectral margin between $A$ and $B$: there exists $\kappa_{\text{spec}} > 0$ such that

$$\langle \mu_A, \mu_B \rangle \leq -\kappa_{\text{spec}}. \tag{79}$$

That is, the spoke means point in sufficiently different directions in spectral space. Then the spoke–spoke mean gap is

$$\delta_{AB} := \|\mu_A - \mu_B\| = \|\mu_A\|^2 + \|\mu_B\|^2 - 2\langle \mu_A, \mu_B \rangle \geq 2(1 + \kappa_{\text{spec}}). \tag{A}$$

**Spoke–hub gap.** By unit normalization, the spoke–hub mean gap satisfies

$$\Delta_{AH} := \|\mu_A - \mu_{H'}\| = 2 - 2\langle \mu_A, \mu_{H'} \rangle \leq 2, \tag{B1}$$

since $\langle \mu_A, \mu_{H'} \rangle \geq -1$. In practice, if $\mu_{H'}$ is a mixture or centroid influenced by both spokes, then $|\langle \mu_A, \mu_{H'} \rangle|$ is not near $-1$, so $\Delta_{AH} \leq C_H < 2$ for some constant $C_H$. To be maximally conservative, one may take $C_H = 2$.

**Ratio bound in spectral space.** Combining equation (A) and equation (B1) yields

$$\frac{\delta_{AB}}{\Delta_{AH}} \geq \frac{2(1 + \kappa_{\text{spec}})}{2} = 1 + \kappa_{\text{spec}} =: c_0 > 0. \tag{C}$$

If a tighter constant $C_H$ is used in equation (B1), replace the denominator 2 by $C_H$ to obtain an even larger $c_0$.

**Plugging into HMH post-filter formula.** From the HSL mean-level derivation in equation (78), we have

$$\frac{\Delta_{AB}}{\Delta_{AH}} = \frac{M}{a+b} \cdot \frac{\delta_{AB}}{\Delta_{AH}} \geq \frac{M}{a+b} \cdot c_0.$$

Because $M \gg a, b$, the right-hand side is strictly greater than 1 (and actually increases with $M$). Therefore,

$$\Delta_{AB} > \Delta_{AH}.$$

So we can say that HSL avoids hub aliasing: the spoke–spoke separation remains larger than spoke–hub overlap, and the ratio improves with scale.

## A.5 THEORM 5

**Case I: Oversquashing:** Oversquashing develops when gradients (or messages) from far nodes shrink exponentially with graph distance, preventing long-range flow of data (Giraldo et al., 2023). The Jacobian of the $L$-layer embedding of node $u$ with respect to the input of node $v$ can be defined as $J_{uv}^{(L)} = \frac{\partial h_u^{(L)}}{\partial x_v}$ where $\delta x_v$ be a perturbation at node $v \in G$ and $\delta h_u^{(L)}$ the change in the output at node $u \in G$. Given the shortest-path distance $d_G(u, v)$, oversquashing occurs if there exist constants $0 < \sigma < 1$ and $D^* \geq 1$ such that

$$\left\| J_{uv}^{(L)} \right\|_2 \leq \sigma^{d_G(u,v)}, \quad \forall L, \ \forall u, v \text{ with } d_G(u, v) \geq D^*.$$

Let the fine-level input features be $h^{(0)} = x \in \mathbb{R}^{N_0 \times d_0}$ with $h_i^{(0)} = x_i$, $i \in V^{(0)}$. For each macro-layer $\ell = 0, \ldots, L - 1$ of the hiearachy we can define the follwing ,

$$\text{Coarsen:} \qquad \tilde{h}^{(\ell+1)} = P^{(\ell+1)} h^{(\ell)} \in \mathbb{R}^{N_{\ell+1} \times d_\ell}$$

$$\text{Haar filter:} \qquad \bar{h}^{(\ell+1)} = \Phi^{(\ell)} \tilde{h}^{(\ell+1)}$$

$$\text{Unpool:} \qquad h^{(\ell+1)} = P^{(0)} \bar{h}^{(\ell+1)},$$

where $P^{(\ell+1)} \in \mathbb{R}^{N_{\ell+1} \times N_\ell}$ coarsens the graph, $\Phi^{(\ell)} = U^{(\ell)} \operatorname{diag}\big(\lambda_{\text{sc}}^{(\ell)}, \Lambda_{\text{wav}}^{(\ell)}\big) U^{(\ell)\top}$ is the (degree-aware) Haar filter, and $P^{(0)} \in \mathbb{R}^{N_0 \times N_{\ell+1}}$ up-samples back to the original nodes. Composing them gives $h^{(\ell+1)} = P^{(0)} \Phi^{(\ell)} P^{(\ell+1)} h^{(\ell)}$. So we define the linear macro-layer map of messages as $M^{(\ell)} = P^{(0)} \Phi^{(\ell)} P^{(\ell+1)}$ where $M^{(\ell)} : \mathbb{R}^{N_\ell \times d_\ell} \to \mathbb{R}^{N_0 \times d_{\ell+1}}$.

With each macro-layer linear,

$$h^{(\ell+1)} = M^{(\ell)} h^{(\ell)}, \qquad \ell = 0, \ldots, L - 1, \qquad h^{(0)} = x, \tag{80}$$

So by unrolling the equation (80) we can write,

$$h^{(1)} = M^{(0)} x, \quad h^{(2)} = M^{(1)} M^{(0)} x, \ldots, h^{(L)} = M^{(L-1)} \cdots M^{(0)} x. \tag{81}$$

Using the chain rule, we can find the change of the message at layer $L$ in terms of feature $X$,

$$\frac{\partial h^{(L)}}{\partial x} = M^{(L-1)} \cdots M^{(0)},$$

hence

$$J_{uv}^{(L)} = \Big[ \prod_{\ell=0}^{L-1} M^{(\ell)} \Big]_{u,v}. \tag{82}$$

After $\ell$ layers of coarsenings, the $u \to v$ effective path has length $k(\ell) \leq \lceil d/r^\ell \rceil$ for some contraction $r > 1$. Assume $u$ and $v$ map to super-nodes connected by a path $i_1, \ldots, i_{k(\ell)} \subseteq V^{(\ell)}$. Define the *path–tube* subspace $T^{(\ell)} := \operatorname{span}\{e_{i_1}, \ldots, e_{i_{k(\ell)}}\} \subset \mathbb{R}^{N_\ell}$ and its pooled image $S^{(\ell)} := P^{(\ell+1)} T^{(\ell)} \subset \mathbb{R}^{N_{\ell+1}}$. By the restricted operator norm we can write (Horn & Johnson, 2012),

$$\|M^{(\ell)}\|_{2, T^{(\ell)}} = \sup_{\substack{x \in T^{(\ell)} \\ \|x\|=1}} \|P^{(0)} \Phi^{(\ell)} P^{(\ell+1)} x\| \geq \sup_{\substack{z \in S^{(\ell)} \\ \|z\|=1}} \|P^{(0)} \Phi^{(\ell)} z\|. \tag{83}$$

**(i) Pool/Unpool stability.** Along the tube, assume the pooling and unpooling are tube-preserving. So, there exist constants $c_p, c_u \in (0, 1]$ ( e.g., minimal soft-assignment mass on the tube) such that

$$\|P^{(\ell+1)} x\|_2 \geq c_p \|x\|_2 \ \forall x \in T^{(\ell)}, \qquad \|P^{(0)} z\|_2 \geq c_u \|z\|_2 \ \forall z \in S^{(\ell)}. \tag{A3}$$

**(ii) Haar-filter passband.** The Haar filter has a scale-local passband floor on $S^{(\ell)}$:

$$\|\Phi^{(\ell)} z\|_2 \geq c_\phi \|z\|_2 \quad \forall z \in S^{(\ell)}, \qquad c_\phi := \inf_\ell \min\big\{\lambda_{\text{sc}}^{(\ell)}, (\lambda_{\text{wav, min}}^{(\ell)})^2\big\} > 0, \tag{84}$$

where $\lambda_{\text{wav, min}}^{(\ell)}$ is the smallest wavelet gain at level $\ell$. The model allows the cross-entropy loss to determine how much the Haar diagonal filter gain should be. If themmodel choose to $\lambda_{\text{wav, min}}^{(\ell)} \geq 1$ then we will have $c_\phi \geq \inf_\ell \lambda_{\text{sc}}^{(\ell)}$.)

Combining equation (83)–equation (84) gives a per-layer tube lower bound

$$\|M^{(\ell)}\|_{2, T^{(\ell)}} \geq c_u c_\phi c_p =: \underline{c} > 0. \tag{A5}$$

Therefore, from equation (82),

$$\|J_{uv}^{(L)}\|_2 \geq \prod_{\ell=0}^{L-1} \|M^{(\ell)}\|_{2, T^{(\ell)}} \geq (\underline{c})^L. \tag{85}$$

After $\ell$ rounds of coarsening, $k(\ell) \leq \lceil d/r^\ell \rceil$, so choosing depth

$$L = \lceil \log_r d \rceil$$

is sufficient to compress the path to $O(1)$. Substituting into equation (85) yields

$$\|J_{uv}^{(L)}\|_2 \geq (\underline{c})^{\lceil \log_r d \rceil} = d^{\log \underline{c}/\log r} \cdot (\underline{c})^{O(1)} \equiv d^{-\alpha} \cdot (\underline{c})^{O(1)}, \qquad \alpha = \frac{-\log \underline{c}}{\log r} \geq 0. \quad (86)$$

The bound of equation (86) is, at worst, a *polynomial* lower bound in the graph distance $d$. But the oversquashing phenomenon would require a *uniform exponential* upper bound $\|J_{uv}^{(L)}\|_2 \leq \sigma^d$ with some $\sigma \in (0, 1)$, for all $L$. A polynomial lower bound cannot be dominated by $\sigma^d$ as $d \to \infty$. Hence, HSL is able to avoid oversquashing.

**Case II: Oversmoothing:** Oversmoothing means different class means collapse for distinct classes (e.g., spokes) $A, B, \|\mu_A^{(L)} - \mu_B^{(L)}\| \to 0$ as depth $L$ grows. In theorem 4 (A.4), we have proved,

$$\frac{\|\mu_A^{(L)} - \mu_B^{(L)}\|}{\|\mu_A^{(L)} - \mu_H^{(L)}\|} \geq \sqrt{M}\left(1 - \frac{2}{\sqrt{M}}\right) > 1, \quad (87)$$

so $\|\mu_A^{(L)} - \mu_B^{(L)}\| > \|\mu_A^{(L)} - \mu_H^{(L)}\|$. Thus, any decay of the spoke–spoke gap forces at least as fast (indeed faster) decay of the spoke–hub gap. Now, we can assume the spoke consists of nodes of the same class, and hubs are the neighbors of the spokes. Equation equation (87) still holds for the assumption. So we can say if over any class means it is separated from the other class, node means irrespective of the neighbor node influence. This ensures that $\lim_{L \to \infty} \|\mu_A^{(L)} - \mu_B^{(L)}\| \neq 0$, avoiding over-smoothing. This completes the proof.

### DATA SOURCE

The PyTorch Geometric `TUDataset` wrapper provided us with access to all the graph-classification datasets used in this work. The TU collection comprises more than 120 datasets, encompassing bioinformatics, chemistry, and social networks. We used the regular versions that come with PyG, unless otherwise specified. We didn't do any additional cleaning of the data.

### A.6 DATASET DESCRIPTIONS

- **PROTEINS.** A biomolecular dataset of 1,113 graphs, with nodes representing protein secondary structure elements (SSEs) and edges indicating proximity in the 3D structure. The graphs are labeled as either enzyme or non-enzyme class. The average graph contains 39.06 nodes and 72.82 edges. The task at hand is binary classification.

- **MUTAG.** A chemical compound dataset consisting of 188 graphs, with nodes representing atoms and edges representing chemical bonds. Graph labels show if a certain bacterium can cause mutations. On average, graphs contain 17.93 nodes and 19.79 edges. Task: binary classification.

- **D&D.** A protein structure dataset consisting of 1,178 graphs, each representing a protein with amino acid nodes and spatially near edges. The label determines if the protein is an enzyme. Graphs are large on average (284.32 nodes and 715.66 edges), which makes scaling this dataset challenging. Task: binary categorization.

- **NCI1 and NCI109.** The National Cancer Institute provides two significant molecular graph databases, each including over 4,000 graphs. Nodes represent atoms, whereas edges correspond to bonds. Labels indicate a compound's activity against non-small cell lung cancer (NCI1) or ovarian cancer (NCI109). Average graph size is $\sim$30 nodes with $\sim$32 edges. Task: binary classification.

- **Mutagenicity.** A large extension of MUTAG with 4,337 molecular graphs. The label indicates a mutagenic effect on a bacterial strain. Average graph size: 30.32 nodes and 30.77 edges. Task: binary classification.

- **IMDB-MULTI.** A dataset of 1,500 ego-networks from the IMDB actor collaboration graph that shows how people work together. Each graph illustrates the connections between performers in the same movie. One of three movie genres is represented by each label. The

average number of nodes in a graph is 13.0, while the average number of edges is 65.94. Task: Classifying into three categories.

- **REDDIT-12K.** A social network dataset with 11,929 graphs representing discussion threads on Reddit. Nodes represent users, edges indicate interaction, and graph labels correspond to the subreddit. The average graph size is large ($\sim$233 nodes and 4,700+ edges), making it a standard benchmark for scalability. Task: multi-class classification.

In Table 4, we described the number of edges, classes, average degree, and average number of edges. We also measured the feature homophily ratio of the datasets using cosine similarity between features. In Figure 5, we showcase the structure homophily of the TU datasets. The cosine homophily is calculated as follows.

**Cosine Homophily.** We compute the cosine-based homophily as:

$$H_{\cos 01} = \frac{1}{|E|} \sum_{(i,j) \in E} \max\left(0, \cos(\mathbf{x}_i, \mathbf{x}_j)\right)$$

where $\cos(\mathbf{x}_i, \mathbf{x}_j)$ is the cosine similarity between node feature vectors $\mathbf{x}_i$ and $\mathbf{x}_j$, and $E$ denotes the edge set of the graph. This measure captures the degree of alignment between the features of connected nodes.

The binary threshold homophily is calculated as follows,

**Binary Threshold Homophily.** We also compute a thresholded binary version:

$$H_{\mathrm{bin}}(\tau) = \frac{1}{|E|} \sum_{(i,j) \in E} \mathbb{I}\left[\cos(\mathbf{x}_i, \mathbf{x}_j) \geq \tau\right]$$

where $\mathbb{I}[\cdot]$ is the indicator function and $\tau = 0.5$ by default. This measure reflects the fraction of edges connecting nodes that are sufficiently similar.

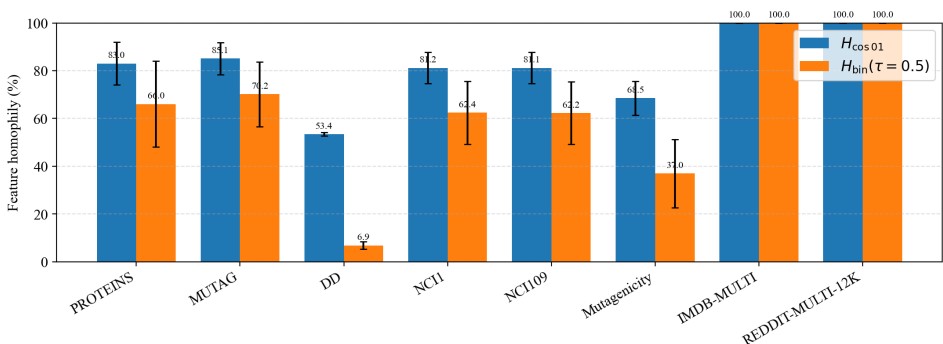

Figure 5: Cosine Feature similarity score for different graph classification datasets

A.3 DATASET STATISTICS

The hyperparameter search is described in detail.

A.7 HYPERPARAMETER SETTINGS

**Shared settings (all datasets).** The following is the description of the hypermeter settings.

- **Optimization.** We trained all models using the Adam optimizer with a learning rate of $3 \times 10^{-3}$ and weight decay of $10^{-4}$ for 100 epochs, reporting test accuracy at the epoch with the best validation performance, while logging training, validation, and test losses and accuracies at each epoch.
- **Geometry.** $L_{\mathrm{mix}} = \alpha_1 L_{\mathrm{topo}} + \alpha_2 L_{\mathrm{feat}}$ with $\alpha = (0.5, 0.5)$; diffusion barrier $M_C$ with strength $\lambda = 0.1$ and final operator $L_C = L_{\mathrm{mix}} + \lambda M_C$. $L_{\mathrm{topo}}$ from the undirected input graph (no added self-loops); $L_{\mathrm{feat}}$ from FAISS $k$NN (IndexFlatL2), Gaussian weights with median–distance bandwidth, symmetrized by $\max(W, W^\top)$.

Table 4: Benchmark datasets from the TU collection. $|\mathcal{G}|$: number of graphs; Avg. $|N|$: average number of nodes; Avg. $|E|$: average number of edges.

| Dataset | $|\mathcal{G}|$ | Classes | Avg. $|N|$ | Avg. $|E|$ | Domain |
|---|---|---|---|---|---|
| PROTEINS | 1,113 | 2 | 39.06 | 72.82 | Biomolecular |
| MUTAG | 188 | 2 | 17.93 | 19.79 | Chemical |
| D&D | 1,178 | 2 | 284.32 | 715.66 | Biomolecular |
| NCI1 | 4,110 | 2 | 29.87 | 32.30 | Chemical |
| NCI109 | 4,127 | 2 | 29.68 | 32.13 | Chemical |
| Mutagenicity | 4,337 | 2 | 30.32 | 30.77 | Chemical |
| IMDB-MULTI | 1,500 | 3 | 13.00 | 65.94 | Social Network |
| REDDIT-12K | 11,929 | 11 | 232.96* | 4,700* | Social Network |

- **Diffusion.** Heat kernel $H_t \approx e^{-tL_{\mathrm{mix}}}$ with $t_{\mathrm{heat}}=0.6$; Chebyshev order 25; diffusion $k$NN via FAISS on the diffusion embedding. Dataset-specific $k_{\mathrm{feat}}$, $k_{\mathrm{diff}}$ (see table).

- **Balanced relaxation.** We estimate the bottom–$K$ *nontrivial* eigenspace of the augmented operator by solving $\min_{U \in \mathbb{R}^{|C| \times K}} \mathrm{Tr}(U^\top L_C U)$ s.t. $U^\top U = I$ and $U^\top s = 0$, where $s \propto D^{1/2}\mathbf{1}$. The orthogonality $U^\top s = 0$ removes the degree-weighted constant mode and enforces degree-balanced partitions (mitigates hub magnetism). In practice we run block power on $B = I - \lambda_{\max}^{-1} L_C$ (12 iters): per step $Q \leftarrow BQ$, project $Q \leftarrow Q - s(s^\top Q)$ to keep $Q^\top s = 0$, then thin QR to re-orthonormalize; the final $U$ (columns of $Q$) approximates the desired bottom–$K$ eigenvectors.

- **Assignments.** Margin-scaled Sinkhorn (temperature $\tau=0.9$, 10 iters) with "cover-all" hardening; switch to margin–softmax by `assign_method="softmax"`.

- **Coarsening.** $X' = S^\top X$, $A' = S^\top AS$; build to the target depth; the classifier uses levels $0..L-2$ (top 1-node level omitted).

- **Haar basis.** Class-aware Haar vectors are propagated down the tree with degree normalization (degree-weighted lifting); the uniform variant is used only in ablations.

- **Classifier.** Per level: linear projection to 32 dims, ReLU, dropout 0.3, learnable diagonal spectral filter; per-level outputs unpooled to level 0 and concatenated; final linear layer outputs logits. Spectral block caps to `max_K` if $K_\ell$ exceeds the cap.

- **Batching / features.** PyG `DataLoader` (dataset-specific batch size); shuffle for train only. If a graph lacks node features, a constant 1-d feature is used.

- **Loss.** $L_{\mathrm{total}} = 0.8\, L_{\mathrm{CE}} + \lambda_{\mathrm{div}} L_{\mathrm{div}}$ with $\lambda_{\mathrm{div}}=0.1$; $L_{\mathrm{rec}}$ is monitored but not weighted in main runs ($\lambda_{\mathrm{rec}}=0$).

- **Compute / reproducibility.** Tree construction on CPU in float64; classifier on GPU if available. Seed $= 42$ for splits and internal randomness.

Table 5: **Per-dataset hyperparameters.**

| Dataset | Levels $L$ | Ratio | $k_{\mathrm{feat}}$ | $k_{\mathrm{diff}}$ | `max_K` | Batch size | Split |
|---|---|---|---|---|---|---|---|
| PROTEINS | 5 | 0.5 | 4 | 4 | 64 | 60 | 80/10/10 |
| MUTAG | 4 | 0.8 | 4 | 4 | 8 | 60 | 80/10/10 |
| D&D | 5 | 0.3 | 8 | 8 | 128 | 16 | 80/10/10 |
| NCI1 | 5 | 0.4 | 6 | 6 | 64 | 64 | 80/10/10 |
| NCI109 | 5 | 0.4 | 6 | 6 | 64 | 64 | 80/10/10 |
| Mutagenicity | 4 | 0.8 | 4 | 4 | 8 | 60 | 80/10/10 |
| IMDB–MULTI | 4 | 0.5 | 4 | 4 | 32 | 128 | 80/10/10 |
| REDDIT–12K | 5 | 0.3 | 8 | 8 | 128 | 8 | 80/10/10 |

## B  NODE CLASSIFICATIONS DATASETS AND EXPERIMENTAL SETUP

**Cora & Citeseer (citation networks).**  Nodes represent publications, and edges represent citation links; features are bag-of-words vectors, and labels are subject areas (7 classes for Cora, 6 for

Table 6: **Ablation across TU graph classification datasets** Mean std over 5 seeds

| Variant | PROTEINS | MUTAG | D&D | NCI1 | NCI109 | Mutagenicity | IMDB-MULTI |
|---|---|---|---|---|---|---|---|
| Full HSL (ours) | $78.5 \pm 0.7$ | $89.5 \pm 0.8$ | $78.2 \pm 0.6$ | $80.1 \pm 0.5$ | $79.8 \pm 0.6$ | $88.9 \pm 0.5$ | $52.4 \pm 1.0$ |
| Topology-only | $76.8 \pm 0.9$ | $87.3 \pm 0.9$ | $76.1 \pm 0.8$ | $78.5 \pm 0.6$ | $78.1 \pm 0.7$ | $86.8 \pm 0.7$ | $50.1 \pm 1.2$ |
| w/o barrier $M_C$ | $77.0 \pm 0.8$ | $88.7 \pm 0.7$ | $76.8 \pm 0.7$ | $78.3 \pm 0.7$ | $78.0 \pm 0.6$ | $88.1 \pm 0.6$ | $49.0 \pm 1.3$ |
| w/o degree-aware Haar | $77.5 \pm 0.7$ | $88.6 \pm 0.8$ | $77.1 \pm 0.7$ | $79.0 \pm 0.6$ | $78.8 \pm 0.7$ | $88.0 \pm 0.6$ | $51.0 \pm 1.1$ |
| w/o balanced relaxation | $76.1 \pm 0.9$ | $88.9 \pm 0.7$ | $75.6 \pm 0.8$ | $77.2 \pm 0.7$ | $77.0 \pm 0.8$ | $88.4 \pm 0.5$ | $48.3 \pm 1.2$ |

Citeseer). We follow the common semi-supervised splits: for Cora, 140 training nodes (20 per class), 500 validation, and 1000 test; for Citeseer, 120 training (20 per class), 500 validation, and 1000 test.

**Actor (co-occurrence network).**  Nodes are actors from Wikipedia; an edge connects two actors that co-appear on a page. Features are keyword counts; labels denote actor categories (5 classes). We use the standard split with 100 training nodes (20 per class), 500 validation nodes, and 1000 test nodes.

**Chameleon & Squirrel (Wikipedia page networks).**  Nodes are pages; undirected edges reflect mutual hyperlinks. Features are keyword indicators; labels partition pages by traffic (5 classes each). We use the "filtered" graphs and the dense splits: 60% train, 20% validation, 20% test.

**WebKB: Texas, Wisconsin, Cornell.**  Nodes represent computer science department webpages, and edges represent hyperlinks. Features are bag-of-words; labels indicate page type (5 classes). We adopt the standard semi-supervised protocol: 20 labeled nodes per class for training, 30 per class for validation, and the remainder for testing.

Table 7: Edge homophily ratio $h$ (fraction of same-label edges). Higher means more homophily. Values from a unified benchmark summary.

| Dataset | Cora | Citeseer | Pubmed | Chameleon | Squirrel | Actor | Cornell | Wisconsin | Texas |
|---|---|---|---|---|---|---|---|---|---|
| $h$ | 0.81 | 0.74 | 0.80 | 0.23 | 0.22 | 0.22 | 0.30 | 0.21 | 0.11 |

*Definition.* We use edge homophily $h(G) = \frac{1}{|E|} \sum_{(i,j) \in E} \mathbf{1}\{y_i = y_j\}$, following prior work. Values in Table 7 are taken from a single source for consistency.[1]

Table 8: **Tree & assignment hyperparameters (per dataset).** $L$: hierarchy depth; Ratio: target cluster ratio per level; $k_{\text{feat}}$: FAISS kNN in feature space; $k_{\text{diff}}$: FAISS kNN in diffusion space; $t$: heat diffusion scale; Cheb: Chebyshev order; $\alpha$: blend in $L_{\text{mix}} = \alpha_1 L_{\text{topo}} + \alpha_2 L_{\text{feat}}$; $\lambda$: barrier strength in $L_C = L_{\text{mix}} + \lambda M_C$; Assign: Sinkhorn parameters; max_K: spectral cap per level.

| Dataset | $L$ | Ratio | $k_{\text{feat}}$ | $k_{\text{diff}}$ | $t$ | Cheb | $\alpha$ | $\lambda$ | Assign ($\tau$/iters) | max_K | Split |
|---|---|---|---|---|---|---|---|---|---|---|---|
| Cora | 5 | 0.50 | 8 | 8 | 0.6 | 25 | (0.5,0.5) | 0.10 | Sinkhorn (0.9/10) | 64 | 20/500/1000 |
| Citeseer | 5 | 0.50 | 8 | 8 | 0.6 | 25 | (0.5,0.5) | 0.10 | Sinkhorn (0.9/10) | 64 | 20/500/1000 |
| Actor | 5 | 0.50 | 8 | 10 | 0.6 | 25 | (0.5,0.5) | 0.10 | Sinkhorn (0.9/10) | 96 | 100/500/1000 |
| Chameleon | 5 | 0.50 | 12 | 12 | 0.6 | 25 | (0.5,0.5) | 0.10 | Sinkhorn (0.9/10) | 128 | 60%/20%/20% |
| Squirrel | 5 | 0.50 | 12 | 15 | 0.6 | 25 | (0.5,0.5) | 0.10 | Sinkhorn (0.9/10) | 128 | 60%/20%/20% |
| Texas | 4 | 0.80 | 4 | 4 | 0.6 | 25 | (0.5,0.5) | 0.10 | Sinkhorn (0.9/10) | 32 | 20/30/*rest* |
| Wisconsin | 4 | 0.80 | 4 | 4 | 0.6 | 25 | (0.5,0.5) | 0.10 | Sinkhorn (0.9/10) | 32 | 20/30/*rest* |
| Cornell | 4 | 0.80 | 4 | 4 | 0.6 | 25 | (0.5,0.5) | 0.10 | Sinkhorn (0.9/10) | 32 | 20/30/*rest* |

## B.1 HEAT KERNEL APPROXIMATION.

To efficiently approximate the heat diffusion operator $\exp(-tL)$ on graphs, we use a truncated Chebyshev expansion, which avoids the need for costly eigendecompositions. Given a normalized

---

[1]Homophily ratios and dataset statistics are from the ICLR'22 study "Is Homophily a Necessity for Graph Neural Networks?", Table 8.

Table 9: **Training & model hyperparameters (per dataset).**

| Dataset | Hidden | Drop | Epochs | Opt (lr/wd) | Loss | Batch |
|---|---|---|---|---|---|---|
| Cora | 48 | 0.50 | 100 | Adam $(3\times10^{-3}/\,10^{-4})$ | $0.8\,L_{\text{CE}} + 0.1\,L_{\text{div}}$ | 1 |
| Citeseer | 64 | 0.60 | 100 | Adam $(3\times10^{-3}/\,10^{-4})$ | $0.8\,L_{\text{CE}} + 0.1\,L_{\text{div}}$ | 1 |
| Actor | 64 | 0.40 | 100 | Adam $(3\times10^{-3}/\,10^{-4})$ | $0.8\,L_{\text{CE}} + 0.1\,L_{\text{div}}$ | 1 |
| Chameleon | 96 | 0.40 | 150 | Adam $(3\times10^{-2}/\,10^{-4})$ | $0.8\,L_{\text{CE}} + 0.1\,L_{\text{div}}$ | 1 |
| Squirrel | 128 | 0.35 | 140 | Adam $(3\times10^{-3}/\,10^{-4})$ | $0.8\,L_{\text{CE}} + 0.1\,L_{\text{div}}$ | 1 |
| Texas | 32 | 0.40 | 100 | Adam $(3\times10^{-3}/\,10^{-4})$ | $0.8\,L_{\text{CE}} + 0.1\,L_{\text{div}}$ | 1 |
| Wisconsin | 32 | 0.35 | 150 | Adam $(3\times10^{-3}/\,10^{-4})$ | $0.8\,L_{\text{CE}} + 0.1\,L_{\text{div}}$ | 1 |
| Cornell | 24 | 0.40 | 140 | Adam $(3\times10^{-3}/\,10^{-4})$ | $0.8\,L_{\text{CE}} + 0.1\,L_{\text{div}}$ | 1 |

graph Laplacian $L$ (with spectrum in $[0,2]$), we shift it to $\tilde{L} = L - I$, so its spectrum lies within $[-1, 1]$, the domain of Chebyshev polynomials.

The heat kernel is then approximated as:

$$\exp(-tL) \approx e^{-t}\left[I_0(t)T_0(\tilde{L}) + 2\sum_{k=1}^{K}(-1)^k I_k(t)T_k(\tilde{L})\right],$$

where $I_k(t)$ are modified Bessel functions of the first kind, and $T_k(\cdot)$ denotes the $k$-th Chebyshev polynomial. This expression is applied to a probe matrix $\Omega \in \mathbb{R}^{n\times r}$, yielding an efficient approximation of $\exp(-tL)\Omega$ without explicitly forming the exponential. In practice, this approximation converges rapidly with a modest number of terms (e.g., $K{=}30$), and is particularly useful in large-scale graph learning.

