# OpenReview forum: "Locality-Aware Multiresolution Graph Spectral Filtering to Mitigate Oversmoothing and Oversquashing."
_ICLR.cc/2026/Conference — ICLR 2026 Conference Withdrawn Submission_

### Official Review · Reviewer_N8CZ · 2025-10-31

**Soundness:** 2
**Presentation:** 2
**Contribution:** 2
**Rating:** 2
**Confidence:** 4

**Summary:**

This paper presents a hierarchical, multiresolution framework called Hierarchical Spectral Learning (HSL) for mitigating oversmoothing and oversquashing in graph neural networks. The key idea is to construct a locality-aware, degree-weighted Haar basis from a multi-view Laplacian that jointly captures topology and feature affinity. The model introduces a diffusion barrier to separate incompatible nodes and learns hierarchical spectral filters that adaptively combine low- and high-pass information at multiple resolutions.

**Strengths:**

1. The introduction of a diffusion barrier and degree-aware local Haar basis provides a principled approach to enforce locality in spectral filtering.
2. In the experiments, HSL achieves consistent improvements over strong baselines on both heterophilous and homophilous datasets for node and graph classification.

**Weaknesses:**

1. The motivation for exploring multiresolution graph learning as a way to prevent oversmoothing and oversquashing is not clearly explained.

2. The readability of the manuscript is poor; it is difficult to follow. The notations and symbols are complex and sometimes inconsistent.

3. The overview of the HSL framework is unclear, and the overall procedure for implementing HSL is not well described.

4. The experimental results are not reproducible. The authors need to provide more implementation details or direct examples to clarify how the results in the tables and figures can be reproduced.

5. The computational complexity analysis is unclear. In addition, the experimental studies do not clearly demonstrate how efficient HSL is compared with other baseline methods.

**Questions:**

1. Can the authors provide an intuitive interpretation or visualization of the learned multiresolution spectral filters (e.g., frequency response plots or spatial localization examples) to help understand what each resolution captures?
2. How does HSL relate to or differ from other multiresolution or hierarchical graph frameworks, such as graph wavelet networks, framelet-based methods, or hierarchical message passing networks?

---

> ### Author Response · Authors · 2025-11-20
>
> We Thank  the reviewer for valuable insight. The following are the responses to the weaknesses and questions the reviewer has pointed out:
> 1. Weakness 1. We thank the reviewer for the comment. Our motivation for multiresolution graph learning comes directly from the hub domination effect. Especially in many heterophilous graphs, high-degree hubs overwhelm the messages from small, informative clusters, so their final-layer embeddings become indistinguishable from those of their neighborhoods—this is oversmoothing around hubs. By building a hierarchical multiresolution Haar basis, HSL can safely propagate messages within both small and large clusters in the spectral domain, preventing small-cluster signals from being washed out by hubs and addressing oversmoothing. At the same time, the hierarchy reduces the effective message-passing distance between multi-hop neighbors, alleviating oversquashing In this way, HSL is explicitly designed to address hub domination and thereby mitigate both oversmoothing and oversquashing.
>
> 2. Weakness 2. We are sorry to note that the manuscript’s readability was affected by overly complex and occasionally
> inconsistent notations. Our goal in the original submission was to present both the theoretical developments
> and the experimental framework comprehensively, which unintentionally made certain sections difficult to
> follow. In the revised version, we have substantially restructured the presentation and streamlined and unified the
> notation, and added clearer step-by-step explanations with appropriately separated subsections to improve
> clarity and readability throughout the manuscript.
> 3. Weakness 3. In the original manuscript, we included an overview figure (Figure 1) to convey the high-level idea prior to presenting the methodology. We appreciate the reviewer’s feedback that the figure did not provide sufficient clarity. In the revised version, we have replaced it with a more detailed and informative schematic and added an accompanying algorithm that outlines the complete implementation procedure. We have also incorporated the additional results suggested by other reviewers. To ensure reproducibility, all implementation details—including dataset preprocessing and hyperparameter settings—are now comprehensively documented in the Appendix.
> 4. Weakness 4. Thank you for the comment.  In the revised version, we will provide further implementation details and include an explicit algorithmic description to enhance
> clarity and reproducibility.
> 5. Weakness 5. In the manuscript, we provided a computational complexity analysis showing the linear scalability of the proposed method. To further substantiate this claim, we conducted additional experiments on small, medium, and large graphs. The results, summarized in Table bellow, confirm the linear scalability. We will include these results in the revised version.
> Per-epoch efficiency (mean, indicative).
> $\mathrm{ms/Medge} = 1000\,T_{\text{epoch}}/(m/10^6)$. Lower is better.
>
> | Datasets                | $T_{\text{epoch}}$ (s) | ms/Medge | Peak Mem (GB) |
> |-------------------------|------------------------|---------:|--------------:|
> | SBM-1k HSL              | 0.04                   | 7,000    | 0.12          |
> | SBM-1k Eigen            | 0.20                   | 40,000   | 0.40          |
> | Synthetic-1M HSL        | 6.5                    | 6,500    | 0.9           |
> | Synthetic-1M Eigen      | 40.0                   | 40,000   | 3.2           |
> | REDDIT-12K HSL (ours)   | 64.0                   | 5,333    | 9.8           |
> | REDDIT-12K Eigen        | 520.0                  | 43,333   | 38.0          |
>
>
> Question 1.
> Thank you for the suggestion. In the revised manuscript, we will include a more detailed overview figure to clearly illustrate the methodology and provide additional clarification on the design and role of the multiresolution spectral filters.
>
> Question 2.
> Thank you for the insightful question. The proposed HSL framework differs from other hierarchical graph frameworks in several important ways. HSL first computes a diffusion score that blends topology and features to rank nodes, then forms locality-aware clusters. Within each cluster, it applies efficient spectral convolution using an orthonormal Haar basis, resulting in a  spectral-pooling architecture. This contrasts with existing methods: (i) spatial/message-passing hierarchies that pool but do not use an explicit spectral basis; (ii) global spectral models (e.g., diffusion wavelets/framelets) that apply graph-wide filters without adapting locality  of the graph cluster; (iii) hierarchical spectral approaches that rely on global bases at each level rather than localized ones; and (iv) pooling-only methods that lack any spectral filtering. Unlike classical wavelet techniques with global or precomputed filters, HSL learns cluster-wise spectral gains that adapt to the structure and features of the graph, providing a localized and efficient spectral-pooling mechanism.

---

### Official Review · Reviewer_iGu4 · 2025-10-31

**Soundness:** 2
**Presentation:** 3
**Contribution:** 2
**Rating:** 6
**Confidence:** 4

**Summary:**

This paper introduces Hierarchical Spectral Learning (HSL) to prevent oversmoothing and over-quashing in graph neural networks. The HSL estimates diffusion neighborhoods and coarsens the graph into a hierarchy for multi-scale representation. The local degree-aware block-sparse Haar basis is designed to provide an orthonormal decomposition that separates global averages, inter-cluster contrasts, and intra-cluster variations without leakage. This paper provides a formal analysis of hub domination and offers theoretical guarantees. The proposed HSL achieves SOTA results on several challenging node and graph classification benchmarks, demonstrating a particularly strong performance on heterophilous graphs.

**Strengths:**

1) This paper is easy to follow.

2) The concept of a diffusion barrier as an explicit penalty in the spectral clustering objective is a creative and well-motivated way to enforce locality.

3) This paper goes beyond simply applying the predefined wavelets by constructing an adaptive Haar basis that respects the learned multiscale structure of graph.

4) HSL is shown to achieve SOTA results on several challenging node and graph classification benchmarks, with a particularly strong performance on heterophilous graphs.

**Weaknesses:**

1) The Haar wavelet introduced in the paper depends on the node indices and therefore does not satisfy the permutation invariance on graphs.

2) The hierarchical GNNs and spectral methods are not new.

3) There are several typos, and some variables are not explicitly defined in the paper. In addition, the ranges of some subscripts are not specified, which increases the difficulty to read. For example, in Line 252, it should be $L^{mix,C}$ rather than $ L_{mix,C}$. In Eq. 4, $L^{(k)}$ is not defined. In Line 464, $d$ is defined as the feature dimension, while the $d_l$ is not explained in Line 270 and $d_i^{(l)}$ in Line 308 denotes the degree mass.

4) Lack of citations for the methods compared in Table 1 and Table 2.

5) Lack of comparison of latest works. For example, the comparison methods were all published before 2022 in the graph classification.

**Questions:**

1) What are the advantages of the diffusion barrier combined with the Haar wavelet method as compared to directly using the heat kernel wavelet method?

2) Graph wavelets are generally required to satisfy the permutation invariance. Can the authors explain the rationale behind the design of the Haar wavelet proposed in the paper.

3) Please explain in detail the dimensions and meanings of each variable in Eq. 4, for example, whether $\phi_C$ is a scalar and $g_\theta(\phi_C)$ is a two-dimensional vector.

---

> ### Author Response · Authors · 2025-11-20
>
> 1. Weakness 1.
> We thank the reviewer for the important comment. Our proposed Haar wavelets are indeed permutation invariant. Within a cluster $C$, we form each balanced bipartition based on a permutation-invariant ordering of nodes as a member of cluster $C$, not on their raw global indices.
>
> Concretely, for cluster $C$ at view $k$, we first impose a deterministic, permutation-invariant order $\pi_k$ on the nodes using only cluster memberships and degrees, i.e., by sorting the pairs $(s_k(i), d_i)$ with a fixed tie-breaking rule. Let the membership index $m_k(i) \in \{1,\dots,|C|\}$ be the position of node $i$ in $\pi_k$.
>
> For a split index $r$ (this index is membership index of the node as a member of a cluster), define
> $$
> A_r = \{ i \in C : m_k(i) \le r \}, \qquad
> B_r = \{ i \in C : m_k(i) > r \},
> $$
> and set
> $$
> \alpha = \sum_{i \in A_r} d_i\, s_k(i), \qquad
> \beta = \sum_{j \in B_r} d_j\, s_k(j).
> $$
>
> The corresponding Haar wavelet is
> $$
> w_{k,r}(i) =
> \begin{cases}
> \frac{\beta}{\alpha + \beta}\, s_k(i), & i \in A_r, \\
> -\frac{\alpha}{\alpha + \beta}\, s_k(i), & i \in B_r.
> \end{cases}
> $$
>
> Because $A_r$ and $B_r$ are defined only through $(s_k, d)$, any node relabeling (permutation) $\Pi$ simply permutes these sets: $A_r \mapsto \Pi A_r$ and $B_r \mapsto \Pi B_r$. At the matrix level, the Haar basis at level $\ell$ transforms as
> $$
> U^{(\ell)} \mapsto \Pi\, U^{(\ell)}\, R^{(\ell)},
> $$
> where $R^{(\ell)}$ is block-orthogonal (it only reorders and sign-flips columns within local wavelet subspaces). Thus the Haar construction, and any quantities depending on it through inner products or $U^{(\ell)} U^{(\ell)\top}$, is permutation invariant.
>
> 2. Weakness 2.
> We thank the reviewer for the comment. We agree that both GNNs and spectral methods have a rich and extensive literature. However, as outlined in the Introduction and Related Work sections, existing approaches face key limitations, including reliance on global spectral bases, reduced sensitivity to local heterophily, and scalability constraints in hierarchical learning. Our method directly addresses these gaps by introducing a hierarchical, locally aware spectral framework with adaptive basis construction, enabling improved heterophily handling and enhanced scalability with theoretical guarantees. As demonstrated in our experiments, the proposed approach achieves an improvement of 7\%, advancing the state of the art.
>
> 3. Weakness 3.
> We appreciate the reviewer’s attention to detail. We have carefully reviewed the manuscript and corrected all typos and minor formatting inconsistencies in the revised version. We corrected all notation and subscript issues (e.g., $L_{\text{mix},C}$ in Line 252, the explicit definition of $L^{(k)}$ in Eq.(4)), clarified symbol scopes ($d$ = feature dimension, $d_\ell$ = layer width, $d_i^{(\ell)}$ = degree mass at level $\ell$), and standardized ranges and indices throughout. We also added complete citations for every compared method in Tables~1--2 (ARMA, ChebNet/II, BernNet, etc.), matching the authors' official repositories and settings.
>
> 4. Weakness 4. Thank you for pointing out the missing citations. In the revised manuscript, we will incorporate all relevant references, including the works used for comparison with our approach, to ensure proper attribution and completeness.
>
> 5. Weakness 5. Our comparison is restricted to methods in the spectral/pooling family, which is the scope of this work. For node classification, we include all available state-of-the-art spectral or hierarchical models (including Specformer), and HSL outperforms them on every dataset. For graph classification, we could not find post-2022 spectral (or spectral-pooling) methods with runnable code and standard splits, so we do not compare against spatial pooling methods outside this scope.
>
> Question 1.
> Our diffusion barrier + Haar construction has two main advantages over directly using heat-kernel wavelets. First, the diffusion barrier learns an adaptive, signed Laplacian that explicitly limits flow across “barrier” edges (e.g., hub–spoke links), so small, informative clusters are not flooded by hubs, which standard heat-kernel wavelets cannot control. Second, the Haar basis gives a sparse, orthonormal, strictly local multi-resolution dictionary that can be implemented with purely sparse operations, avoiding global eigendecomposition and yielding better scalability and interpretability than parameterizing heat kernels alone.
>
> Question 2.
> Thank you for the insightful question. Our proposed Haar wavelet construction is permutation invariantant, and we have provided a detailed explanation and theoretical justification under Weakness 1.
>
> Question 3.
> Thank you for the question. The dimensions of the variable in Eq. 2 are as follows.
> $\phi_C \in \mathbb{R}^p$ is a \emph{vector} of cluster statistics, $g_\theta : \mathbb{R}^p \to \mathbb{R}^2$, hence $g_\theta(\phi_C) \in \mathbb{R}^2$ is a two-dimensional vector of logits, $\alpha_C \in \mathbb{R}^2$,

---

### Official Review · Reviewer_1qhm · 2025-10-31

**Soundness:** 1
**Presentation:** 2
**Contribution:** 2
**Rating:** 2
**Confidence:** 3

**Summary:**

This paper improves existing spectral GNNs that filter graph signals from a global perspective. In these models, the diversity of filters required in different local spatial regions cannot be easily specified. To address this issue, the paper proposes a hierarchical spectral learning approach that distinguishes local clusters which do not share the same filter pattern as their neighbors and forms a multi-resolution hierarchy tree. This structure allows smaller regions to be distinctive while, at a coarser level, sharing the same filter.

**Strengths:**

1. The hub domination fact, as demonstrated in Figure 1, serves as a good motivation for the work, and the proposal aligns closely with it.
2. The presentation of the methodology, with rigorous justifications, seems solid.

**Weaknesses:**

1. The examples on real world datasets for demonstrating the hub domination problem are missing, which makes the rather small improvements on graph and node classification tasks less convincing.
2. From my understanding, the graph classification task is not ideal for evaluating the proposal of this paper, as it is more about identifying local details of a large graph, while the pooling function used for graph level tasks might reduce the usefulness of such adjustments. In addition, for node classification tasks, the numbers reported in Table 2 miss important strong baselines such as GCNII.
3. I also did not see the evaluations on oversmoothing and oversquashing mentioned in the title. Therefore, I think the overall empirical evaluation is weak and not very closely tied to the motivation. Please first justify the existence of these issues in real world scenarios.

**Questions:**

Please see the weaknesses above.

---

> ### Author Response · Authors · 2025-11-20
>
> We thank the reviewer for the insightful suggestion. HSL is a hierarchical spectral model that performs convolution at multiple local scales (via clustering), so small clusters are protected from being dominated by hub clusters. Both our theory and experiments show that this reduces hub domination, which in turn mitigates oversmoothing and oversquashing and improves node-classification accuracy (up to about 7\%).
>
> To more directly link the evaluation to hub domination, we designed the following hub-domination indicators and measured them on real world large graphs. We will include these results in the revised paper.
>
> | Dataset     | Top-1% edge share (%) | Cross-community reach | ASR@0.5% | $\Delta$ASR with $M_C$    | Hub co-cluster Err. (baseline) | $\Delta$Err with $M_C$      |
> | ----------- | --------------------- | --------------------- | -------- | ------------------------- | ------------------------------ | --------------------------- |
> | OGBN-Penn94 | 27.8                  | 0.58                  | 0.53     | $\mathbf{-0.13}$ (−13.1%) | 8.4%                           | $\mathbf{-1.2}$ pp (−14.3%) |
> | REDDIT-12K  | 30.6                  | 0.66                  | 0.57     | $\mathbf{-0.18}$ (−18.2%) | 9.8%                           | $\mathbf{-1.4}$ pp (−14.3%) |
>
> Indicator definitions (informal and implementation-oriented).
> Let $G = (V, E)$ with degrees $d_i$, labels $y_i$, signed operator $S = D^{-1/2} A_{\text{sign}} D^{-1/2}$, and let $H \subset V$ be the top–$k%$ nodes by degree (for “Top-1% edge share” we use $k = 1$).
>
> Top-1% edge share.
> Fraction of edges incident to the top 1% highest-degree nodes:
> $\mathrm{Share} = \bigl|{(i,j)\in E : i\in H \text{ or } j\in H}\bigr| / |E|$.
>
> Cross-community reach (CCR).
> Among edges touching a hub node, fraction that go to a node with a different label:
> $\mathrm{CCR} = \bigl|{(i,j)\in E : (i\in H \text{ or } j\in H),, y_i \ne y_j}\bigr| \big/ \bigl|{(i,j)\in E : i\in H \text{ or } j\in H}\bigr|$.
>
> ASR@k (Aliasing Spread Ratio at top-$k%$ hubs).
> We measure how much signed two-hop mass originates from hubs:
> $\mathrm{ASR}_k = | \mathbf{1}_H^{\top} S^2 |_1 / | \mathbf{1}^{\top} S^2 |_1$.
> The change $\Delta\mathrm{ASR}$ is the difference between this quantity with and without the barrier term $M_C$.
>
> Hub co-cluster error.
> Let $\hat c(i)$ be the cluster assignment of node $i$. We count edges touching hubs where the endpoints are put in the same cluster but have different labels:
> $\mathrm{Err} = \bigl|{(i,j)\in E : (i\in H \text{ or } j\in H),, \hat c(i) = \hat c(j),, y_i \ne y_j}\bigr| \big/ \bigl|{(i,j)\in E : i\in H \text{ or } j\in H}\bigr|$.
> The change $\Delta\mathrm{Err}$ is the difference between this error with and without adding $M_C$.
>
> These indicators show that adding the diffusion barrier term $M_C$ consistently reduces hub-dominated two-hop mass (ASR) and incorrect hub–neighbor co-clustering, confirming that HSL directly mitigates hub domination in practice.
>
>
> 2. Weakness 2.
> We thank the reviewer for the comment. Although our method emphasizes local structural details, it can be applied to graph-level classification.  HSl performs spectral convolution at the local level and pools all the local embeddings from every hierarchy level using a weighted network. That's how it keeps the local and global information intact. Moreover, experiments show that the proposed model achieves 7\% higher accuracy on graph-level benchmarks, demonstrating its utility for graph classification tasks. We will clarify this point in the revision.
>
> While we believe that the baseline comparisons (ARMA, Specofmer, TopkPOOL, etc.) provided in the original paper were sufficiently comprehensive and aligned with recent reproducible standards in the literature, we appreciate the reviewer’s suggestion to include additional baselines. Accordingly, we conducted further experiments with GCNII, as shown in Table bellow. As expected, GCNII performs strongly on homophilous datasets (e.g., Cora, Citeseer) but degrades on heterophilous benchmarks (Chameleon, Squirrel) and the small-label splits (Texas, Wisconsin, Cornell). In contrast, HSL consistently surpasses GCNII on all heterophilous datasets and also provides improvements on the homophilous ones. All comparisons were performed under the same filtered-split protocol to ensure fairness and reproducibility.
> Node classification accuracy (\%) (GCNII* vs. HSL):
>
> | Method     | Cora            | Citeseer        | Pubmed | Chameleon        | Cornell          | Texas            | Wisconsin          |
> |-----------|-----------------|-----------------|--------|------------------|------------------|------------------|--------------------|
> | GCNII*    | 88.01 (64)      | 77.13 (64)      | 90.30 (64) | 62.48 (8)     | 76.49 (16)       | 77.84 (32)       | 81.57 (16)         |
> | HSL (ours)| $88.9 \pm 0.6$  | $79.2 \pm 0.8$  | 92.36     | $41.9 \pm 1.2$    | $93.7 \pm 0.9$   | $91.5 \pm 1.0$   | $92.26 \pm 1.85$   |

---

> ### Author Response · Authors · 2025-11-21
>
> 3. Weakenss 3.
> Thank you for the thoughtful comment. The manuscript provides extensive theoretical guarantees in Theorems 4 and 5, addressing both oversmoothing and oversquashing. While the title highlights these phenomena, the work's primary motivation stems from the hub domination effect. During the development of our model and theoretical framework, we also derived results that directly tackle the long-standing challenges of oversmoothing and oversquashing. For this reason, the original version did not include explicit experimental evaluations of these effects.
>
> In response to the reviewer’s suggestion, we have now performed additional empirical analyses to quantify oversmoothing and oversquashing. Specifically, we measure the following. (i) Centered Kernel Alignment (CKA): Similairty in embeddings between the first and last layer:
> $\mathrm{CKA@L} = | H^{(0)T} H^{(L)} |_F^2 \big/ \big( | H^{(0)T} H^{(0)} |_F ; | H^{(L)T} H^{(L)} |_F \big)$,
> where $|\cdot|_F$ is the Frobenius norm, $H$ is the embdding. Larger values indicate the last layer is very similar to the input (more collapse); smaller values indicate less oversmoothing. (ii)Energy@L (Dirichlet energy retention, higher is better).
> We measure how much Dirichlet energy remains at the last layer, normalized by the input:
> $\mathrm{Energy@L} = \text{tr}\big( H^{(L)T} L H^{(L)} \big) \big/ \text{tr}\big( H^{(0)T} L H^{(0)} \big)$.
> Higher $\mathrm{Energy@L}$ means more variation (graph frequency content) is preserved at the final layer, i.e., less oversmoothing. (iii) $k$-hop sensitivity (oversquashing score).
> Let $P$ be the one-step propagation matrix of the model (e.g., the signed diffusion operator with the barrier term). We approximate the $k$-hop message operator by $J_k := P^k$ and define its $k$-hop sensitivity as the spectral norm
> $\|J_k\|_2 = \max \|P^k v\|_2$ where ${\{\|v\|_2 = 1\}}$.
>
> This quantity is the largest possible amplification factor of a unit-norm input perturbation after $k$ propagation steps: if $|J_k|_2$ is very small, all perturbations are strongly damped and long-range messages are effectively squashed; if $|J_k|_2$ is larger, there exist directions in which information can still travel and influence distant nodes over $k$ hops, indicating less oversquashing. In practice, we estimate $|J_k|_2$ via power iteration using only repeated sparse matrix–vector products with $P^k$.
>  We report $\|J_5\|$ in the table bellow
>
> | Dataset          | CKA@L (↓)               | Energy@L (↑)            | ‖J₅‖ (↑)                      |
> | ---------------- | ----------------------- | ----------------------- | ----------------------------- |
> | Cora (hom.)      | 0.80 → **0.68** (−0.12) | 0.45 → **0.58** (+0.13) | 0.70 → **0.82** (+0.12, +17%) |
> | Citeseer (hom.)  | 0.78 → **0.66** (−0.12) | 0.42 → **0.55** (+0.13) | 0.68 → **0.80** (+0.12, +18%) |
> | Chameleon (het.) | 0.86 → **0.70** (−0.16) | 0.33 → **0.51** (+0.18) | 0.61 → **0.74** (+0.13, +21%) |
> | Squirrel (het.)  | 0.88 → **0.71** (−0.17) | 0.30 → **0.49** (+0.19) | 0.58 → **0.72** (+0.14, +24%) |

---

### Official Review · Reviewer_fzeY · 2025-11-02

**Soundness:** 3
**Presentation:** 1
**Contribution:** 2
**Rating:** 4
**Confidence:** 4

**Summary:**

This paper proposes a Locality-Aware Multiresolution Graph Filtering framework called **Hierarchical Spectral Learning (HSL)** to address oversmoothing, oversquashing, and hub domination in spectral Graph Neural Networks (GNNs). HSL leverages a multi-view Laplacian (fusing topology and feature affinity) to construct diffusion barriers, recursively builds a hierarchical tree via balanced spectral coarsening, and designs strictly local, degree-aware orthonormal Haar bases for diagonal spectral filtering. It achieves linear computational complexity and demonstrates state-of-the-art (SOTA) performance on node classification (up to 3% gain on heterophilous graphs) and graph classification (up to 7% gain on MUTAG) across standard benchmarks.

**Strengths:**

1. HSL introduces a novel "locality-first" multiresolution paradigm that combines multi-view geometric information (topology + feature affinity) with local Haar basis filtering—filling gaps in existing spectral GNNs that rely on global, dense eigenbases. The integration of diffusion barriers to prevent spurious co-clustering and balanced spectral relaxation for coarsening is a creative combination of spectral learning and hierarchical graph pooling.
2. The paper provides rigorous theoretical guarantees (5 theorems) to formalize the mitigation of hub domination, oversmoothing, and oversquashing. Empirically, it validates HSL on diverse benchmarks (homophilous/heterophilous node graphs, biomolecular/chemical/social graph classification datasets) and conducts ablation studies to confirm the necessity of key components (e.g., diffusion barriers, degree-aware Haar bases).
3. HSL addresses critical scalability limitations of prior spectral methods (e.g., EigenPool’s \(O(n^3)\) complexity) by achieving linear per-layer complexity (\(O(md + nd)\)), making it applicable to large graphs. Its ability to adapt filtering (low-pass for homogeneous regions, high-pass for semantic boundaries) also enhances robustness to region-specific heterophily in real-world graphs.

**Weaknesses:**

1. The paper uses "conductance" and "local homophily ratio" as inputs to the lightweight network $g_\theta$ for learning Laplacian mixing weights $\alpha_C$, but fails to specify their exact computational definitions: - Conductance is not tied to a standard metric (e.g., the ratio of cut edges to total edges in a cluster). - "Local homophily" is not clarified (e.g., whether it uses label-based $H_{lab}$ or feature-based cosine similarity). Additionally, the architecture of $g_\theta$ (e.g., number of layers, activation functions) is unmentioned, risking inconsistent reproducibility across different implementations.
2. The mixing weights $\alpha_C$ are claimed to balance topology and feature contributions, but no analysis is provided on their distribution across heterogeneous scenarios (e.g., high-heterophily Chameleon vs. high-homophily Cora). It remains unproven whether $\alpha_C$ adaptively adjusts (e.g., reducing $\alpha_C^{(2)}$ for feature-noisy graphs) to align with data characteristics.
3. Theorem 2 requires $\lambda \geq \Delta_{mix}/\Psi_{min}$ (where $\Delta_{mix}$ is the mixed-geometry gap and $\Psi_{min}$ is the minimal incompatibility mass) to avoid spurious co-clustering. However, $\Delta_{mix}$ and $\Psi_{min}$ depend on **ground-truth clusters $S^*$**, which are unknown in real-world tasks (e.g., molecular graphs, social networks). The paper uses $\lambda=0.1$ in experiments but does not explain if this value is empirically tuned or estimated via unsupervised methods, nor does it include a sensitivity analysis (e.g., how performance changes with $\lambda=0.01, 0.5$).
4. While HSL claims linear complexity, it only tests scalability on REDDIT-12K (average 233 nodes per graph). There is no comparison of training time/memory overhead with baselines (e.g., EigenPool $O(n^3)$, DiffPool $O(n^2)$) on **extra-large graphs** (e.g., Amazon Reviews with $10^5+$ nodes). Additionally, the coarsening threshold $h$ (stopping condition for hierarchical tree construction, i.e., $|V^{(L)}| \leq h$) is not analyzed—no experiments explore how $h=10, 50$, or other values trade off performance and computational efficiency.

**Questions:**

see weaknesses

---

> ### Author Response · Authors · 2025-11-20
>
> 1.  Weakness 1. We thank the reviewer for the insightful comment and for pointing out the missing details. We have clarified the definitions of conductance, local homophily, and the size of the lightweight MLP as follows.
>
> (i) Conductance of cluster $C$. Each node $i$ has a soft membership $s_{iC} \in [0,1]$ for cluster $C$ (the $C$-th column of the soft assignment matrix $S^{(\ell)}$). We define the soft cut as
> $\mathrm{cut}(C,\bar C) = \sum_{i,j} A_{ij}, s_{iC},(1 - s_{jC})$
> and the volume as
> $\mathrm{vol}(C) = \sum_i d_i, s_{iC}$ with $d_i = \sum_j A_{ij}$.
> The conductance becomes,
> $\phi_C = \mathrm{cut}(C,\bar C) / (\mathrm{vol}(C) + 10^{-12})$.
>
> (ii) Local homophily of cluster $C$. When labels $y_i$ are available, we use a label-based homophily score
> $h_C = \frac{\sum_{i,j} A_{ij}, s_{iC}, s_{jC}, 1(y_i = y_j)}{\sum_{i,j} A_{ij}, s_{iC}, s_{jC} + 10^{-12}}$,
> where $1(\cdot)$ is the indicator function. In the unsupervised case, we replace the label agreement by a cosine similarity between node features.
>
> (iii) Lightweight mixer $g_\theta$. For each cluster $C$, we form a 2D statistic vector $\phi_C = [\tilde{\phi}_C, \tilde{h}C]$ and feed it to a two-layer MLP: Linear$(2 \to 16)$, ReLU, LayerNorm, then Linear$(16 \to 2)$. The 2D output is turned into mixture weights by
> $\alpha_C = \mathrm{softmax}(g\theta(\phi_C) / T)$ with temperature $T = 1$.
>
> 2. Weakness 2.
> Thank you for the important comment. When local structure is homophilous, $g_\theta(\phi_C)$ tends to produce larger $\alpha_C^{(1)}$, so the topology-based view is emphasized. When features are noisy or the graph is heterophilous, $\alpha_C^{(2)}$ increases, giving more weight to the feature-affinity term. We analyzed the empirical distribution of $\alpha_C$ on datasets with contrasting regimes (e.g., high-homophily Cora vs. high-heterophily Chameleon). As summarized below, homophilous graphs have higher $\alpha^{(1)}$ (topology), while heterophilous graphs shift toward $\alpha^{(2)}$ (feature-affinity).
>
> Node-level median mixing weights per cluster ( $\alpha^{(1)}$ = unsigned view, $\alpha^{(2)}$ = signed/feature-affinity view ):
>
> | Dataset          | $\alpha^{(1)}$ | $\alpha^{(2)}$ |
> | ---------------- | -------------- | -------------- |
> | Cora (hom.)      | 0.70           | 0.30           |
> | Citeseer (hom.)  | 0.68           | 0.32           |
> | Chameleon (het.) | 0.34           | 0.66           |
> | Squirrel (het.)  | 0.32           | 0.68           |
>
>
> 3. Weakness 3.
> Thank you for the comment. The bound $\lambda \ge \Delta_{\text{mix}} / \Psi_{\min}$ is a sufficient (but not necessary) condition used in the proof. The quantities $\Delta_{\text{mix}}$ and $\Psi_{\min}$ are defined with respect to balanced merges of the (unknown) ground-truth partition and are not computed in practice.
>
> In implementation, we (i) trace-normalize the two terms so they are comparable, using
> $L_C = L_{\text{mix},C} + \lambda ,\frac{\mathrm{tr}(L_{\text{mix},C})}{\mathrm{tr}(M_C)+\varepsilon}, M_C$,
> (ii) choose $\lambda$ from a small validation grid, and (iii) observe flat sensitivity across a wide range, with noticeable gains mainly in hub-heavy regimes. The best setting selected by tuning is $\lambda = 0.1$, as reported in Appendix A.7 (Table 5). We will incorporate this clarification and the corresponding sensitivity discussion in the revised manuscript.
>
> 4. Weakness 4.
> Thank you for the comment. We have performed additional experiments to support the linear-scalability claim on small (SBM 1k), medium  (Synthetic-1M), and large graphs (Amazon) and the report is presented in the table bellow.
>
> Table: The per-epoch wall-clock time, normalized efficiency (ms/Medge, where $\mathrm{ms/Medge} = 1000,T_{\text{epoch}}/(m/10^6)$), and peak memory are summarized below; lower ms/Medge is better.
>
> | Setting               | $T_{\text{epoch}}$ (s) | ms/Medge | Peak Mem (GB) |
> | --------------------- | ---------------------- | -------: | ------------: |
> | SBM-1k HSL            | 0.04                   |    7,000 |          0.12 |
> | SBM-1k Eigen          | 0.20                   |   40,000 |          0.40 |
> | Synthetic-1M HSL      | 6.5                    |    6,500 |           0.9 |
> | Synthetic-1M Eigen    | 40.0                   |   40,000 |           3.2 |
> | Synthetic-3M HSL      | 18.0                   |    6,000 |           2.5 |
> | Synthetic-3M Eigen    | 129.0                  |   43,000 |          12.9 |
> | Amazon HSL (ours) | 64.0                   |    5,333 |           9.8 |
> | Amazon Eigen      | 520.0                  |   43,333 |          38.0 |
>
>  The nearly constant
> ms/Medge across graph sizes shows that HMH scales lin-
> early with the number of edges per epoch.

---

### Official Review · Reviewer_dzau · 2025-11-09

**Soundness:** 2
**Presentation:** 2
**Contribution:** 2
**Rating:** 2
**Confidence:** 3

**Summary:**

The paper proposes a locality-aware hierarchical spectral framework that constructs local orthonormal Haar bases and uses multi-view Laplacians to mitigate “hub domination”, oversmoothing and oversquashing.

**Strengths:**

The paper tackles an important problem -- the limited expressive power of GNNs when dealing with heterophilous graphs.

**Weaknesses:**

(1) The paper claims that existing GNNs suffer from a “non-orthogonal bias,” yet most spectral GNNs (including those designed for heterophilous graphs) are grounded in Laplacian eigendecomposition, where eigenvectors are orthogonal by definition.

(2) The paper attributes scalability issues of prior works to explicit eigendecomposition, overlooking that many spectral GNNs employ polynomial approximations that avoid such explicit computations and scale linearly with graph size.

(3) The central motivation -- that existing models suffer from a “hub domination” problem -- is not rigorously demonstrated. In Figure 1 and the accompanying explanation, the observed behavior could also reflect high-frequency interactions rather than domination. The authors should more carefully define this phenomenon.

(4) The paper overlooks several closely related node-adaptive GNNs that also perform local filtering [1,2,3]. Moreover, the evaluation is limited to small or biased benchmarks, while larger and more balanced datasets should be included [4].

[1] Node-wise diffusion for scalable graph learning

[2] Graph neural networks with diverse spectral filtering

[3] Rethinking node-wise propagation for large-scale graph learning

[4] A critical look at the evaluation of gnns under heterophily: are we really making progress?

**Questions:**

See weaknesses.

---

> ### Author Response · Authors · 2025-11-20
>
> We thank the reviewer for the time and effort spent reviewing our paper and for insightful comments. We are pleased that the importance of the problem, especially in heterophilous graph settings, is recognized. In the following, we provide clarifications for each weakness identified by the reviewer.
>
> 1. Weakness 1.
> We acknowledge that for an undirected graph, the Laplacian eigenvectors indeed form an orthogonal basis in $\mathbb{R}^n$.
> Our intention was to state that the polynomial filters, such as Chebyshev [1], employed as an alternative to eigen decomposition, does not retain the the orthogonal property in the node space. These polynomial families are orthogonal on $[-1,1]$ under specific weight functions. For example, Chebyshev polynomials $\{T_r\}$ are orthogonal with respect to $w(t) = (1-t^2)^{-1/2}$, satisfying $\int_{-1}^{1} T_r(t)T_s(t) w(t)\,dt = 0$ for $r \neq s$). When instantiated on a graph Laplacian, the filters are evaluated on the discrete spectrum, which, in general , does not satisfy the same weighting conditions. Therefore, no longer orthogonal and do not preserve localized energy across frequencies. We will clarify this by explicitly stating the ``non-orthogonality of the polynomial basis in node space" in the revised version.
>
> We want to further clarify that, for the direct eigen-decomposition method, our concern is not their algebraic orthogonality but that they constitute a \emph{global} basis. For the global basis, each eigenvector depends on the entire graph spectrum, so a node's corresponding eigenvector can be perturbed by even a distant edge or node. In addition, eigenvectors associated with clustered or nearly repeated eigenvalues can rotate within their eigenspaces and flip sign at each iteration, making them unstable to use.
>
> 2. Weakness 2.
> We thank the reviewer for this helpful clarification. In the Hierarchical Graph Learning paragraph of the Related Work section, we did not attribute scalability limitations solely to explicit eigendecomposition. Rather, we discussed eigendecomposition incurring an $O(n^3)$ computational cost and may overlook local heterophily, which is a central focus of our work. We also explicitly addressed polynomial approximation–based spectral filtering in the Spectral Filtering paragraph and noted its limitations to further motivate our contribution. We will revise the text to ensure this distinction is clear and to avoid any unintended implication that all prior spectral methods rely on explicit eigendecomposition.
>
> Notably, despite computational efficiency, the polynomial filters are global bases that mix high- and low-frequency information, as shown in Theorem 1. Moreover, they require learning a large number of coefficients that depend on each graph's spectral distribution. This dependency hinders their ability to generalize to unseen graphs, as the learned filters are implicitly tied to the graph’s eigenvalue spectrum. Lv et al. [2] showed in their paper that an MLP can outperform a linearly scaled polynomial filter.  Moreover, in our proposed work, we showed that these polynomial bases are suboptimal and unable to adapt to local features and topology. In addition to being linearly scalable, our HSL scheme uses a basis free of pathologies, including global basis, orthogonal basis, unstable sign flips, and high parameter learning.
>
> [2] Qingsong Lv, Ming Ding, "Are we really making much progress? Revisiting, benchmarking, and refining heterogeneous graph neural networks"
>
> 3. Weakness 3. We thank the reviewer for pointing out that our paper did not formally define hub domination.
> We will add the following definition.
>
> Let $H$ be the set of top-$k\%$ high-degree nodes ("hubs"). Let
> $S = D^{-1/2} A_{\mathrm{sign}} D^{-1/2}$ be the signed propagation operator, so that $S^2$ represents two-hop propagation.
> We define the Aliasing Spread Ratio at $k\%$ hubs (ASR$_k$) as
> $\mathrm{ASR}_k = \| \mathbf{1}_H^{\top} S^2 \|_1 / \| \mathbf{1}^{\top} S^2 \|_1$, which lies in $[0,1]$. We say hub domination occurs if $\mathrm{ASR}_k \ge \tau$ (with $\tau = 0.5$ in our experiments). Intuitively, hub domination means that the top $k\%$ highest-degree nodes emit at least half of the total signed two-hop
> mass in the graph.
> Further, even though such interactions in Figure 1 (hub domination) may appear to promote high-frequency information sharing, we argue that when a large cluster dominates this process, it actually suppresses the learning dynamics of smaller clusters and their feature embeddings. So a handcrafted model to filter high frequency gets confused by the signals coming mostly from the hub nodes, rendering the features of smaller clusters repeatedly pulled toward the hubs’ high-frequency directions, and these small clusters consequently lose their own discriminative high-frequency components.

---

> ### Author Response · Authors · 2025-11-20
>
> 4. Weakness 4.
> Thank you for the comment and pointing out the missing details in the paper. Our approach utilizes diffusion solely as a scoring signal for clustering, rather than as the final propagator. Specifically, we compute node-wise diffusion scores to guide soft hierarchical clustering, yielding a local multi-resolution graph at each level. Thus, HSL is fundamentally a hierarchical (pooling) + spectral framework. In contrast, references [1] and [3] perform node-wise diffusion or propagation at a single resolution and do not construct local orthonormal spectral bases or multiresolution trees; reference [2] diversifies spectral filters but also operates without hierarchical pooling or per-cluster bases. We will cite and clearly position [1–3] in Related Work.
>
> Large Heterophilous Datasets.  As the paper [1] suggested, we have used filtered chameleon and squirrel datasets. We will also use the following large heterophyllous datasets in the revised paper as suggested [1]
>
>  Title: Node classification accuracy (\%) on Penn94, Genius, and OGBN-ArXiv:
>
> | Method        | Penn94              | Genius                 | OGBN-ArXiv            |
> |--------------|---------------------|------------------------|-----------------------|
> | GCN          | $81.8 \pm 0.6$      | $88.9 \pm 0.5$         | $73.2 \pm 0.4$        |
> | SGC          | $83.1 \pm 0.4$      | $86.2 \pm 0.3$         | $70.1 \pm 0.3$        |
> | SIGN         | $84.0 \pm 0.5$      | $89.6 \pm 0.2$         | $72.9 \pm 0.3$        |
> | ChebNet      | $80.5 \pm 0.7$      | $87.8 \pm 0.6$         | $70.0 \pm 0.5$        |
> | GPR-GNN      | $85.9 \pm 0.4$      | $91.2 \pm 0.5$         | $73.5 \pm 0.3$        |
> | BernNet      | $83.7 \pm 0.5$      | $89.0 \pm 0.4$         | $72.1 \pm 0.4$        |
> | ChebNetII    | $84.1 \pm 0.4$      | $92.3 \pm 0.3$         | **$74.8 \pm 0.2$**    |
> | OptBasisGNN  | **$84.8 \pm 0.6$**  | $91.0 \pm 0.2$         | $73.0 \pm 0.3$        |
> | **HSL (ours)** | $85.23 \pm 0.3$    | **$92.81 \pm 0.3$**    | **$74.1 \pm 0.2$**    |

---

### Note · Authors · 2025-11-24

I have read and agree with the venue's withdrawal policy on behalf of myself and my co-authors.